# Integrating climate change induced flood risk into future population projections

Evelyn G. Shu [1] ✉, Jeremy R. Porter[1,2,3], Mathew E. Hauer [4], Sebastian Sandoval Olascoaga[5], Jesse Gourevitch[6,7], Bradley Wilson[1,7], Mariah Pope [1,7], David Melecio-Vazquez [1,7] & Edward Kearns [1,7]

Flood exposure has been linked to shifts in population sizes and composition. Traditionally, these changes have been observed at a local level providing insight to local dynamics but not general trends, or at a coarse resolution that does not capture localized shifts. Using historic flood data between 2000-2023 across the Contiguous United States (CONUS), we identify the relationships between flood exposure and population change. We demonstrate that observed declines in population are statistically associated with higher levels of historic flood exposure, which may be subsequently coupled with future population projections. Several locations have already begun to see population responses to observed flood exposure and are forecasted to have decreased future growth rates as a result. Finally, we find that exposure to high frequency flooding (5 and 20-year return periods) results in 2-7% lower growth rates than baseline projections. This is exacerbated in areas with relatively high exposure to frequent flooding where growth is expected to decline over the next 30 years.

Much of the world's population is exposed to some kind of extreme weather event exacerbated by climate change[1]. These events have been directly connected to impacts on human systems including economic, social, and political crises[2]. Adding to the problem's complexity, humans have concurrently settled in areas with climate risk that were previously thought to have low, or no, exposure[3]. Since 1980, there have been over 300 disaster events responsible for $2.3 trillion in damages across the US[4]. Nearly half of these damages can be attributed to flooding and tropical cyclones[4]. As such, understanding the ways that flood risk impacts human systems is of great importance.

Increasing flood exposure and losses are expected to drive population and demographic shifts in the US. Some projections estimate that globally, up to 216 million people may migrate due to climate change by 2050[5]. It is now generally accepted that sea level rise (SLR) could lead to large migrations as impacted communities move away from rising flood risk[6–9]. Previous research has documented an observed association between such flood exposure and the shifting demographics of an area as drivers of an emerging phenomenon termed climate gentrification[10]. It is worth noting that at a global scale, these types of population shifts are much more common and are most often linked to involuntary migration due to climate exposure[11] or, more colloquially, climate migration[12].

Past research on climate migration related to flood risk often focuses on relatively long-distance relocations, such as across county borders[6,8,13–15]. This existing research has found that areas with high climate risk (or exposure) are currently seeing an influx of population and growing[16–21]. However, increasing evidence suggests that a more voluntary form of climate migration is associated with flood risk at a more local scale, taking into account property and neighborhood level variations in exposure[10,22,23]. Much of this research is focused on the

[1]First Street Foundation, Brooklyn, NY, USA. [2]Department of Sociology and Demography, City University of New York, New York, NY, USA. [3]Environmental Health Sciences Department, Columbia University's Mailman School of Public Health, New York, NY, USA. [4]Department of Sociology and Center for Demography and Population Health, Florida State University, Tallahassee, FL, USA. [5]Department of Urban Studies and Planning, Massachusetts Institute of Technology, Cambridge, MA, USA. [6]Environmental Defense Fund, San Francisco, CA, USA. [7]These authors contributed equally: Jesse Gourevitch, Bradley Wilson, Mariah Pope, David Melecio-Vazquez, Edward Kearns. ✉e-mail: evelyn@firststreet.org

stressors of lower impact but persistent flood events rather than the less likely shock from extreme flood events[24].

Climate migration is likely to increase over time, with projections suggesting between 4.2 and 13.1 million people in the U.S. may be at risk of inundation from sea level rise by 2100, and suggests that in the absence of adaptation there are likely to be large migrations on a similar scale to the Great Migration in the twentieth century[25], during which 4 million people may have moved[26]. With 21.8 million properties currently having some exposure to flooding within the Contiguous United States (CONUS), and 23.5 million expected to have flood exposure by the middle of the century[27], it is important to understand how migration and settlement patterns may change over time[6,28–30].

When climate impacts are evaluated over longer time periods, it is likely that a combination of direct (e.g. property inundation) and indirect (e.g. neighbors moving away) factors will contribute to migration[6,31]. In addition, these migrations are likely to occur along existing pathways[15,28,32], thus likely constituting an enhanced normal out-migration[6]. Such expectations are driven by migration theory, which draws heavily on the economics of location through an evaluation of factors referred to as push and pull factors[33]. In the case of climate migration, the emphasis is generally put on the push factors as they relate to exposure and movement away from an area e.g[11]. However, the lack of such exposure may also serve as a pull factor[34].

To understand climate risk, it is important to clarify the definitions of risk components: hazard, exposure, and vulnerability. In this context, hazard refers to the probability and severity of physical environmental events. Exposure is concerned with the physical proximity to those hazards, capturing the degree to which assets, people, and systems are exposed. Finally, vulnerability refers to the susceptibility of an exposed system to the impacts of a hazard, often encompassing various social, economic, or political factors that influence the ability to cope with a hazard event[35,36]. Below, the focus of this research highlights that a flood exposure metric is the primary differentiating factor used in the modeling framework.

Within that context, much existing research is limited by the availability of high-quality flooding information to help define at risk populations. Within the U.S., national scale flood information is often limited to historical records or flood maps provided by government agencies, while high-resolution flood risk information is often only locally available. In addition, the effect of adaptation on flood risk is considerable[37], and built adaptation should be accounted for as they generally exist in areas at the intersection of high levels of exposure and population concentration. Furthermore, high-resolution flood risk information for multiple return periods (RPs) and sources (pluvial, fluvial, and coastal) allow for a more complex investigation of how populations react to different flood severities, frequencies, and sources. For example, past research has shown that communities tend to rebound after a devastating but low-likelihood event, but the degree of recovery varies by demographic and socioeconomic characteristics[13,14]. Other research has shown that frequent and less severe flooding is correlated with more persistent issues of decreased property valuations and population redistribution in favor of higher ground[10,38].

Many studies looking to estimate displaced populations due to flooding rely on an underlying assumption that exposure corresponds directly with migration. Estimates for SLR displacement are highly variable due to differing types of exposure relating to who is at risk of migration. Common at risk definitions include populations (1) in low-elevation coastal areas, (2) in 100-year floodplains, and (3) predicted to be inundated in the future under SLR (Hauer et al.[7] provides a general discussion of these approaches–which typically prioritize future understandings of coastal flooding, with less, or no, emphasis on inland flooding). However, in addition to inundation, there are multiple other important aspects to consider. Those considerations revolve

around the fact that, depending on the context, residents will react differently to the same level of flood exposure. In addition, there may be external differences related to the manifestation of impacts (i.e. indirect vs. direct), the existence of amenities, and the simple fact that heightened risk exists in some of the most amenity heavy areas of the US. For example, many low-lying areas within 100-year floodplains possess ample water resources, thus potentially attracting people to live there for reasons that correlate with the same factors (flooding) that may be used to inform out-migration[39]. Research that equates inundated populations to the migration population may be overly simplistic if they do not adequately include other covariates in their modeling.

While there is some understanding that populations respond differently to severe and rare flooding, compared with more frequent flooding, this research has focused on single events and investigations of singular types (sources, severities, or probabilities) of flooding. For example, while Hauer[6] contributed towards understanding inland population impacts due to flood-related migration, the source of flood information used was still solely from SLR inundation. Advancements in agent based modeling (ABM) are moving towards modeling household level migration under climate change[40–42], but it has been limited to flooding increases from SLR and requires strong assumptions and extensive existing research to define inputs (for example, amenity values must be previously defined, and thus in these applications are limited), and are computationally intensive due to the use of individual flood simulations. In addition, much of this previous research forecasting future growth in flood exposure uses independent projections of populations and flooding, and does not investigate the historical relationship between the two to modify future population projections[19–21].

In this work, we estimate localized shifts in population counts (2000–2020) across small area administrative units (specifically, Census Blocks) across the CONUS, accounting for exposure from multiple flood sources, depths, and probabilities. This research is explicitly focused on the migration component of population change and has taken care to include covariates known to be related to differential levels of fertility and mortality in the research design, captured by social, economic, and spatial factors. In addition, the research design explicitly removes any changes in fertility and mortality by focusing only on 2 decades of data. Given the possible endogeneity of the treatment (flood exposure), we use propensity score matching to balance unobserved covariates that affect both the treatment and the outcome and synthetically create a control-treatment design. This is then followed by a statistical modeling process that isolates the historical flood-population relationship. Building upon that retrospective analysis, a secondary component of this research projects population levels out 30 years in the future, relying on existing population projections along Shared Socioeconomic Pathways (SSPs)[43], climate-adjusted flood hazard projections[44,45], and historically derived relationships between flood exposure and population redistribution.

## Results

### Future national level population projections adjusted for historic climate impact

The future population projections indicate that when introducing the climate consequences derived from the modeling process, there are overall negative impacts on population growth in areas with high flood exposure. The climate consequence across the country is evaluated in Table 1, where the population projections for blocks which are defined as relatively highly exposed to flooding (in respect to three different indicators, defined in the following sentence) are shown to decrease, or have decreased growth, when compared to their respective baseline projections. Blocks were categorized into dichotomous groups for comparison based on the following definitions: if a block has higher

**Table 1 | Integration of climate consequence into future population projections**

| 5, 20, or 100-year Flood Exposure | Mean diff. |
|---|---|
| % change in number of people in treatment blocks vs SSP baseline | 14.02 |
| % change in number of people in treatment blocks vs SSP baseline + climate consequence | 12.47 |
| % Diff: Treatment, SSP baseline vs. Treatment Growth, SSP baseline + climate consequence | −1.54 |
| **20-year Flood Exposure** | |
| % change in number of people in treatment blocks vs SSP baseline | 9.95 |
| % change in number of people in treatment blocks vs SSP baseline + climate consequence | 8.55 |
| % Diff: Treatment, SSP baseline vs. Treatment Growth, SSP baseline + climate consequence | −1.40 |
| **5-year Flood Exposure** | |
| % change in number of people in treatment blocks vs SSP baseline | 5.76 |
| % change in number of people in treatment blocks vs SSP baseline + climate consequence | −1.42 |
| % Diff: Treatment, SSP baseline vs. Treatment Growth, SSP baseline + climate consequence | −7.19 |

Table shows breakdown of future population growth rates for baseline populations, projected populations (integrated climate consequence), and the difference between the two. These rates are shown with different dichotomous group breaks based on the following definitions: if a block has higher than average proportion of inundated properties in the (1) 5, 20, or 100-year RPs, (2) 20-year RP, and (3) 5-year RP.

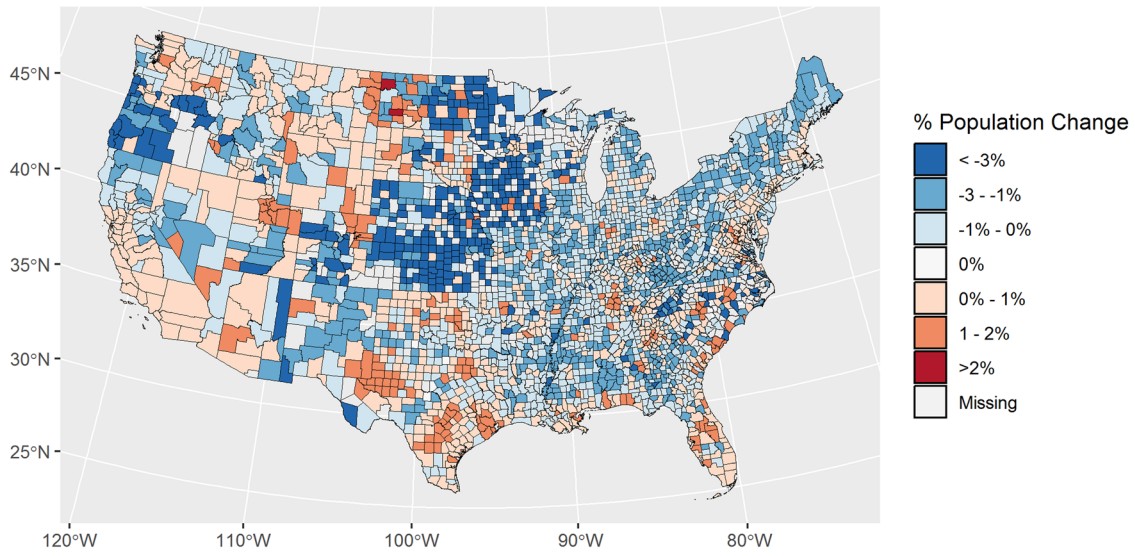

**Fig. 1 | County level projected population change resulting from the application of the climate consequence to the SSP2 future population projections.** Population change is shown as an annualized rate, comparing projected future population counts (~2053) to current population counts (~2023). Spatial layers were obtained from the U.S. Census Bureau.

than average proportion of inundated properties in the (1) 5, 20, or 100-year RPs, (2) 20-year RP, and (3) 5-year RP.

Table 1 highlights that areas with exposure to more frequent flooding report a more pronounced negative climate consequence associated with the integration of observed population responses from historic flood risk. Specifically, blocks with relatively high exposure in the 5-year RP report a negative 7 point difference in the rate of growth in the baseline projections (5.7% cumulative growth) versus those corrected by the integration of the climate consequences (1.5% cumulative loss). Similarly, results show that communities with high inundation in the 20-year RP will also grow at slower rates than the current SSP population projections estimate with a 1.4 point loss on the cumulative rate of growth in the baseline projections (10%) versus the cumulative rate of growth in the climate adjusted projections (8.6%). Finally, the national number consolidating the impact of all RPs is collectively negative with about a 2 point difference in the cumulative rate of growth in the baseline population projections (−14%) versus the cumulative growth in those same areas following the integration of the climate consequence (−12%).

**Current and future variations in absolute population exposed**
Figure 1 shows the spatial distribution of the percentage of future population changes that is due to the integration of the modeled climate consequence. The differentiation is interesting to note because, while there is a climate consequence associated with the tipping points identified in this research, there are also larger macro level drivers of population change that are built into the baseline SSP2 trajectory used as the baseline for this forecast. Again, given the highly localized nature of flood risk, the acute impact needed to be explored at a higher resolution.

In order to look at these results at a higher resolution, the block level population forecasts were examined for select areas known to have high, and uneven, levels of flood exposure in both the coastal and inland context. One such area due to coastal flooding is Miami-Dade County, FL, presented in Figs. 2 and 3. A second example of inland flooding, Cincinnati, OH, is presented and discussed for comparison in the Supplementary Information. These examples illustrate the integration of the modeled climate consequence into future population projections at a localized level. At a summary level, Miami-Dade shows

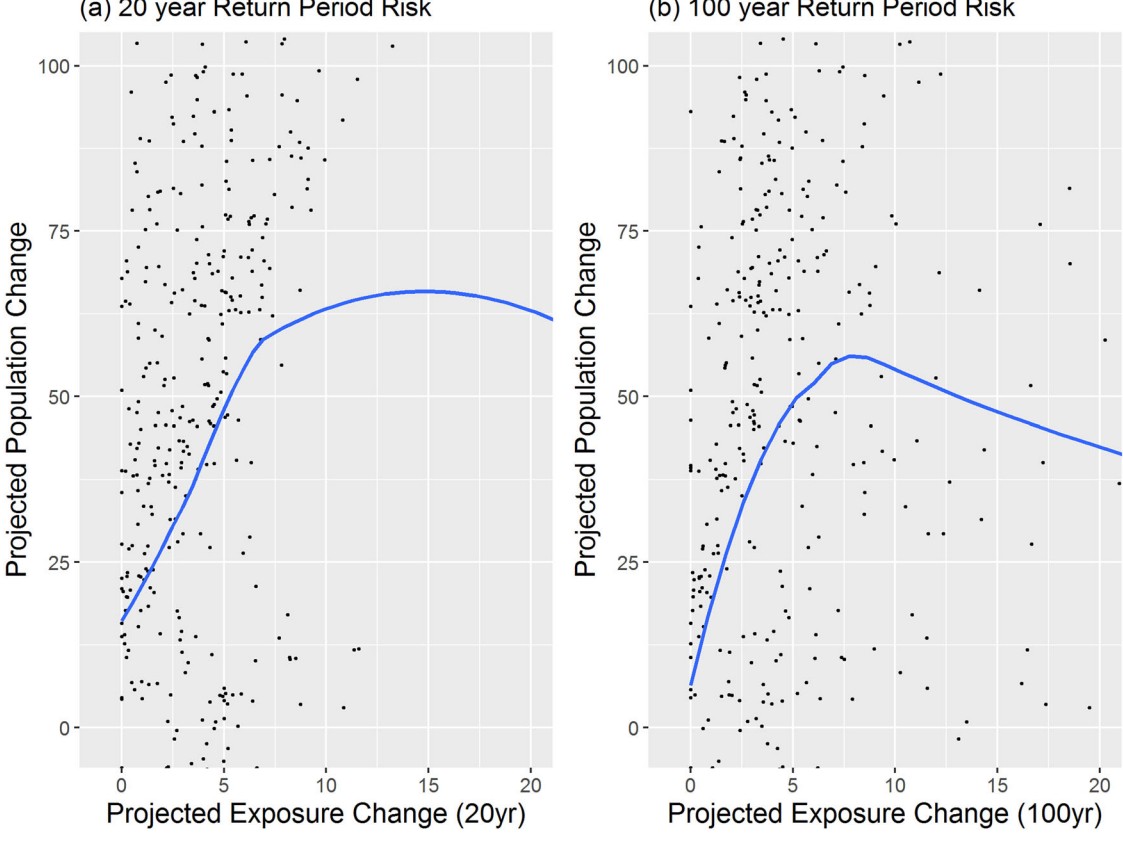

**Fig. 2 | Example of future flood exposure and population change.** Relationship between exposure within the (**a**) 20 and (**b**) 100-year RP with population change, future forecasted population change between 2023 –2053 (Miami-Dade County, FL). Data are presented as mean values, with a randomized sample of future.flood exposure and population counts.

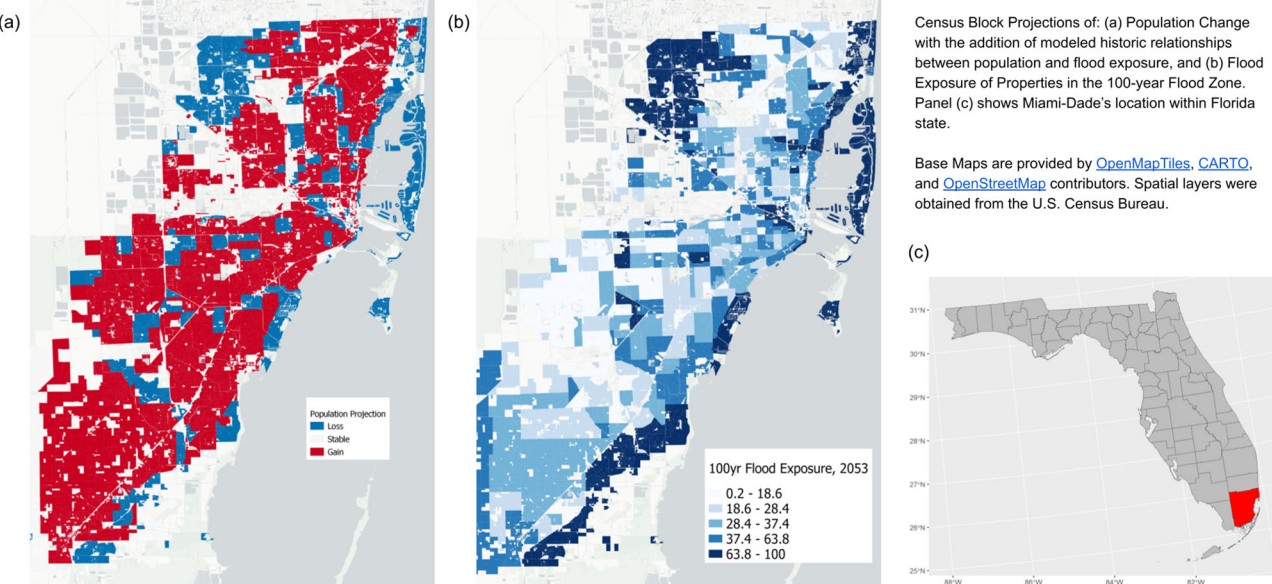

**Fig. 3 | Projected population change resulting from the application of the climate consequence to the SSP2 future population projections (Miami-Dade, FL).** Census Block Projections of: (**a**) Population Change with the addition of modeled historic relationships between population and flood exposure, and (**b**) Flood Exposure of Properties in the 100-year Flood Zone. Panel (**c**) shows Miami-Dade's location within Florida state. Base Maps are provided by OpenMapTiles, CARTO, and OpenStreetMap contributors. Spatial layers were obtained from the U.S. Census Bureau.

that the inclusion of flood exposure coefficients in future projections are related to a negative effect on population growth. However, it should be noted that the relationship is not specifically linear in either case. Instead, there appears to be a tipping point across the different RPs as the proportion of properties that are exposed in the census block reaches a point in which population change begins to decline based on the empirically derived relationships. In contrast, an opposite overall effect exists in Cincinnati, OH where areas of higher inundation

are actually more likely to grow versus areas less exposed. A discussion of potential reasons behind this inland pattern is further discussed in the Supplementary Information around Supplementary Fig. 6.

Figure 2 highlights the pattern of this relationship for all blocks in the city of Miami-Dade in the 20 and 100-year RPs. In this case, population change actually increases up to about 7–10% of the properties in the area which have exposure to flooding in that RP. However, after 7–10% tipping point, there is a negative relationship associated with population decline. The pattern is most pronounced in the 100-year exposure for all the reasons related to frequency and severity above.

To understand the relationships identified in Fig. 2, it is important to locate the blocks spatially. Figure 3 highlights the relationship between forecast population change and changing flood exposure (by selected RP scenarios) out to mid-century. While the county predictions indicate that Miami-Dade is likely to grow into the middle of the century, that growth will be uneven and it is highly correlated with the exposure identified in the 100-year flood zone. This highlights that when seeking to understand the future impact of flood exposure change, using high precision data in regard to both flood exposure and population change, are required to see an adequate picture of the impacts. In particular, this example shows the areas along the Atlantic Ocean, Miami Beach, and the areas closest to the low-lying Everglades as the areas most susceptible to increased risks of flooding and to the resulting population decreases that are forecasted to occur as a result.

## Discussion

This research presents a method for investigating observed relationships between historic flood exposure and population redistribution patterns in an attempt to understand the impact of climate change on population change forecasts. When isolating that effect from other migration push-pull factors, we are left with an independent, mostly negative, relationship over the past 20 years. Using that empirically derived relationship, this research further showed that summary level analyses at the state and county levels hide the spatial variability associated with the localization of climate exposure. However, when looking at this data at a high resolution, we see clear patterns associated with changing flood exposure and its impacts on estimated future population levels. This holds true for both coastal and inland areas, although the effects are more dramatic in coastal areas.

This analysis shows that while at an aggregate level, areas with high climate risk may be growing in population, this growth is more complicated, at least in regard to flood risk. Specifically, while larger geographic areas with high flood exposure may be growing rapidly, when looking within those areas at a higher resolution, the results found here indicate that growth is on average slower or negative for areas that are relatively more exposed than other areas in the same state. By accounting for population and spatial characteristics, migration likelihoods may be captured in population growth (and decline) rates as associated with various sources, depths, and probabilities of flood exposure at a high resolution. This not only captures varying responses to flood exposure, but also captures the population effect for areas that have relatively less flood exposure than surrounding areas (where this lack of flooding serves as a "pull" factor).

The intention of this manuscript is to investigate the influence of relatively high levels of exposure on population growth rates, the model variables which distinguish between relatively high and low exposure implicitly also captures the population effect for areas that have relatively less flood exposure than surrounding areas, where this lack of flooding serves as a pull factor. While current adaptation is accounted for in the FSF-FM used in this analysis, one limitation of the analysis is that it does not forecast future adaptation which may serve to dampen push factors or exacerbate pull factors. This serves as a potential source of uncertainty in future projections, and provides opportunities for future research. In addition, there may be uncertainty at multiple other steps throughout this process. The first forms

of uncertainty come from the underlying flood, population, and other input data. Historical data may have uncertainties due to accuracy and measurement issues, while future data (flood and population) may have additional uncertainties as it builds off of historical data and uses various modeling approaches, capturing climate uncertainties, modeling uncertainties, and uncertainties due to underlying assumptions.

In spite of these known issues, this manuscript illustrates the potential for high-resolution data in providing a better understanding of climate migration within the U.S. While some areas with a high level of exposure have continued to see growth and are predicted to continue growing into the future, when data are used which provides information at a localized level, a more complex story comes to light. While larger areas (i.e. counties) with high exposure have grown and may continue to grow, small (i.e. tracts) areas with relatively low exposure may be driving this growth. A more clear understanding of how and where these impacts may occur will allow for better decision making as communities prepare for the impacts of climate change and continue to provide insight into how individuals are dealing with the changing climate and increasing levels of flood risk.

## Methods

We integrate data on flood exposure and population with a series of indicators related to local political, social, and economic conditions (Supplementary Table 1). Historic flood exposure is accounted for through the data detailed in Lai et al.[46], and from the NOAA Storm Events Database[47]. In addition to the historical flood exposure data, the First Street Foundation Flood Model (FSF-FM) estimates of properties with exposure from the underlying 3-meter hazard model were also included for the 5-, 20-, 100-, and 500-year RPs. These exposure estimates account for fluvial, pluvial, and coastal sources and are used to produce the primary treatment variable in this analysis, the proportion of properties in a block that are exposed to flooding. Both historic exposure and current estimates of the risk of exposure were included to account for two separate dimensions of risk associated with the experience of flooding and the likelihood that one could flood, thus accounting for prior sorting based on experiences of actual flooding. Future flood information is also provided by the FSF-FM, projecting flood exposure out 30 years into the future under RCP 4.5 using a combination of 21 global climate models, holding all else constant except for climatic changes[44,45].

As this manuscript uses the proportion of inundated properties, we refer to the flood information as exposure for the remainder of the manuscript. More broadly, other variables which are accounted for in this research include demographic, economic, and political factors, which capture additional aspects of vulnerability, thus constituting the necessary components of risk[35]. These additional vulnerability-related metrics are not operationalized in such a way to provide a uniform comparison of risk across areas (such as by constructing a risk index), so the use of the term risk is kept only in broad contexts.

Future baseline population projections, which are demographically driven and do not explicitly account for climate change impacts at a local level, are provided by Hauer[43] and are proportionally downscaled using current block population counts. This study utilizes SSP2 for consistency with the flood data future projections along RCP4.5 (equivalent to SSP2). SSP2 is defined as the Middle of the Road scenario, where future development trends are similar to historical patterns, and there is slow progress in achieving sustainable goals[48]. Additional data sources are introduced in Supplementary Table 1 and the following section highlights additional data sources and the larger 3-step research design.

**Step 1. Propensity matching for control-treatment pairs by flood exposure**
To account for potential endogeneity due to factors which may influence self-selection into block areas with flood risk, propensity scoring

was executed to synthetically create conditions akin to a control-treatment design. Several options for treatment variables were considered, where a binary variable was used to define considerable flooding versus not-considerable flooding in an area. In this analysis, considerable flooding is a within-state metric that was derived from sensitivity analyses to identify a census block for the treatment group if that census block has a proportion of inundated properties that is ~3 times the mean proportion of inundated properties within census blocks in the same state. The metric captures relative flood exposure, and a block in a highly exposed state would need to have higher flood exposure to be considered at risk. This threshold was selected to retain a large sample size while also prioritizing the inclusion of highly exposed blocks. Conditional on observable characteristics of the block areas, propensity score matching allows for the synthetic construction of comparable treatment and control groups within a sample, allowing for a more direct comparison of the impacts from flooding on population trends and reducing selection bias. The variables used for the propensity score matching are presented in Supplementary Table 1. These variables are also more broadly related to some key drivers in differences between population growth rates, including job density, job growth rates, population in the starting year (2000), population density, median income, median home value, the Multi-Deprivation Index (MDI; provided by the Census Bureau, capturing six dimensions of standard of living, health, education, economic security, housing quality, and neighborhood quality[49]), and employment rate. A matching algorithm is used where matches are selected using the nearest neighbor as determined by a generalized linear model[50]. Furthermore, because migration tends to be within local areas, and generally within labor/housing markets, the subsequent analyses are run independently for each state across CONUS[51].

## Step 2. Evaluating the historic impact of flood exposure on population trends

The propensity score matching process resulted in a within-state matched sample, which is then fitted independently for each state through an OLS estimator, to account for political factors which may vary considerably across states, while not requiring models to rely on small and overly homogenous samples.

The dependent variable of our models is the block change rate, where:

$$block\ change = (population\ t_{i,t+n} - population\ t_{i,t})/ \\ (population_{i,t} \times (1+n)) \tag{1}$$

The historic block change is calculated for 2000 to 2020. The subscripts on the above equation include $i$ for blocks, and $t$ for time. The modeling process then combined the explanation of variation in the calculated block change metric with the local amenities and/or disamenities. The modeling approach was further optimized to only select conditions that were empirically identified to be related to the explanation of block change variation within the state-level models. In all cases, the full model could include indicators of flood exposure, social characteristics, economic characteristics, local area pull amenities, spatial location, and baseline future change (from the unmodified baseline projections, as a latent variable of unobservable amenities). As such, when considering the explanatory modeling of the variation in the block change, the full relationship that the model may choose from may be expressed as the following equation:

$$block\ change = b_0 + Flood + Social + Economic + Amenities \\ + Spatial + future\ change + \varepsilon \tag{2}$$

And the minimum relationship that the model must include is:

$$block\ change = b_0 + Flood + \varepsilon \tag{3}$$

Where *Flood* within the minimum model refers to a set of variables capturing the proportion of inundated properties within a block level. These variables exist for various all 4 RPs to capture differential population responses to variable flood scenarios, where the responses in redistribution patterns after low-severity and high-frequency flood events may differ from reactions to high-severity and low-frequency flood events. While data from the FSF-FM does include 2-year RP flooding (to measure tidal flooding), it is difficult to isolate it from other coastal amenities without meaningfully reducing the sample size. As such, this modeling approach does not explicitly account for nuisance tidal flooding events, but instead operationalizes frequent events at the 5-year RP.

In the full model, additional *Flood* variables include the square of the proportion of inundated properties for all RPs to allow for non-linear relationships. *Social* captures a set of variables, including initial population counts and density. *Economic* includes variables such as job opportunities, incomes, home valuation, employment rates, and a multidimensional deprivation index score. *Amenities* includes distances to natural amenities associated with flood exposure, such as proximities to coasts or rivers. *Spatial* is a set of variables that capture some of the spatially explicit interactions and movements of population settlements, such as longitudes, latitudes, and county controls. *Future change* is a single variable which is provided by the baseline growth projections, aimed at capturing any other characteristics of an area that facilitate certain growth rates. Both the spatial and future change variables also account for differences in differing valuation views by populations, such as perceptions of flood risk which may influence decision-making[53], suitability of neighborhoods for raising a family, and social connectivity. For many of these variables, linear, interaction terms, and exponential variations (accounting for any non-linear relationships) were included in the full model selection for the final model to consider. The model selection process was optimized to select the most efficient model based on AIC. Here, the minimum and full model specifications were provided, where the model iterates through all possible variations between the two to select the one with the lowest AIC, capturing both goodness-of-fit and model simplicity. This process allows for the inclusion of location and population-specific valuation views, which are difficult and costly to independently quantify.

To evaluate the efficacy of the forecasting methodology, the data are split into training and testing sets through a stratification process. For each state, 80% of the total sample is used to construct the training sample, and the remaining 20% serves as the testing sample set. To evaluate the accuracy of the trained model in making predictions for the testing dataset, three accuracy metrics are considered[25]. Given the evaluation of these accuracy metrics, as well as the conclusions that they are all absolutely low and don't seem to have any systematic patterns, the methodology to estimate coefficient estimates was deemed reliably useful in this application. This validation process and the results are discussed in further depth in the Supplementary Information (Supplementary Tables 2–3, Supplementary Figs. 1–2).

## Step 3. Integrating the climate consequence to Future SSP Population Projections

For the purposes of this study, we modify population projections only with the condition of changes in flood exposure. To do this, we apply the coefficients for flood variables to the future (2053) flood layer projections assuming operationally linear increases between current

and future flooding projections and that this gradual increase in exposure will be realized through growth rates. The coefficients for the relationship between flood variables and block growth rates may be applied through the integration of these relationships. Where initially the future population projections are derived as age-sex-race adjusted demographic estimates along the SSP2 trajectory.

$$future\,growth_{baseline}(annual) = demographic\,indicators \quad (4)$$

Where, expressed differently using future (baseline) population counts, may be considered as:

$$future\,growth_{baseline}(annual) = \frac{future\,pop_{baseline} - pop_{present}}{pop_{present}} \quad (5)$$

Similarly, the future projections correcting for flood exposure (i.e. integrating the climate consequence) may be thought of as the contribution of and relationship between flood exposure metrics and demographic indicators,

$$future\,growth_{projected}(annual) = demographic\,indicators \\ + flood\,indicators \quad (6)$$

Where expressed differently using future (projected, with flooding indicators described in step 2) population counts, may be considered as:

$$future\,growth_{projected}(annual) = \frac{future\,pop_{projected} - pop_{present}}{pop_{present}} \quad (7)$$

Where, when combined the future population projections can be expressed as:

$$future\,pop_{projected} = (demographic\,indicators + flood\,indicators) \\ \times pop_{present} + pop_{present} \quad (8)$$

The retrospective analysis and prospective correction of population projections along the SSP2 trajectory give us the ability to take observable associations between flood risk and exposure and integrate a high resolution climate signal into current SSP projections. Most importantly, we are interested in how areas that have been classified as highly exposed to flooding are projected to change in population relative to areas that have been classified as not being highly exposed.

### Reporting summary
Further information on research design is available in the Nature Portfolio Reporting Summary linked to this article.

## Data availability
The input datasets used for this analysis are either already publicly available or cannot be made available due to restrictive data sharing agreements. All data sources are listed with links to access points in Supplementary Table 1.

## Code availability
The code used for this analysis is available in a Github repository at: https://github.com/FirstStreet/flood-risk-migration-projection.

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

## Acknowledgements

The views expressed in this paper are solely the responsibility of the authors and should not be interpreted as reflecting the opinions of any organizations from which public data was used.

## Author contributions

E.G.S. and J.R.P. conceptualized and designed the research project, assembled input data, analyzed output data, and wrote the manuscript. E.G.S. designed and ran the statistical models. J.R.P. supervised the project. M.E.H. and S.S.O. provided advice on model design, analyzed output data, and edited the manuscript. J.G., B.W., M.P., D.M.V. and E.K. provided advice on the analysis of output data and edited the manuscript. J.G., B.W., M.P., D.M.V. and E.K. all contributed equally to this work.

## Competing interests

The authors declare no competing interests.
