## [Peer Review File · Nature Communications]

Integrating Climate Consequences from Flood Risk into Future Population ProjectionsREVIEWER COMMENTS

Reviewer #1 (Remarks to the Author):

In this paper, the authors start by developing a “hyper local” model of the association between historical and current flood risk and exposure and population change in the United States, followed by using the estimated coefficients to project population change out to 2055. One of the main findings overall is that the association between flooding and population change is slightly negative (i.e., more/higher flooding is associated with population decline); however, as the authors rightly point out, this finding obscures significant local heterogeneity. In what follows, I provide my major comments, questions, and concerns in an effort to help the authors improve this paper and, ultimately, their contribution to the literature.

In general, I thought the framing of this paper was well organized and complete with the exception of at least two trouble spots. One, at the top of p. 4, is that the authors need to more clearly introduce, discuss, and distinguish the concepts of exposure and risk, as well other related concepts such as vulnerability. The second, which occurs at a couple of points, is that the authors sometimes vacillate between saying “migration” and “population change.” While, in an ideal world, they would have the ability to focus in on migration, in the actual world, they can only get at population change. So, more care is needed not only with respect to consistent word choice, but also with bringing in consideration of the other two components of population change—namely, fertility and mortality.

Another question that arose for me throughout this paper is how to think about the authors’ “hyper local” approach in light of their reliance on estimating average effects that are then organized at the level of states. For instance, on p. 9, the authors state, “there are overall negative impacts on population growth in areas vulnerable to flood risk,” However, upon reading this, I caught myself wondering whether an overall negative effect really means anything, especially in light of some of the authors’ later comments about spatial heterogeneity and maps such as those in Figure 7. To the authors credit, the authors do dig into specific cases such as Miami-Dade County, FL. However, in doing so, I think they reinforce the false choice between taking an average effects one-size-fits-all approach and a case specific approach. As a middle ground, I suggest that the authors consider splitting the difference and taking a comparative case study approach that balance case-specific heterogeneity and causal attribution.

Reviewer #2 (Remarks to the Author):

Comments to the authors

The manuscript “Integrating Climate Consequences from Flood Risk into Future Population Projections” presents a modelling study estimating population levels in 2055, whilst accounting for exposure to flood hazard. The methodology presented in this study is applied to census blocks within the contiguous United States. The study first assesses the effect of current flood risk on population development from 2000 to 2020. These findings are then used to adjust a population projection under Shared Socioeconomic Pathway 2 (SSP2). The study finds that current flood risk affects population development in areas of high exposure, resulting in uneven growth in for example Texas, Florida, and California. Population growth is expected to decline and reverse in areas at disproportionate risk within the coming 30 years.

This works presents interesting findings and could be a relevant contribution to the literature on this topic. However, the several issues need to be addressed for the research to qualify for publication.

General comments:

1) Although the researchers mention “climate consequences” the manuscript title, the research only uses historic flood hazard and does not account for changes in flood risk under climate change. Assuming flood hazard will remain the same in the future is an assumption that should be stated explicitly. If the authors do account for climate change, the authors should specifically state which

climate change scenarios (or RCP) are analyzed.

2) The authors mention the use of 'several flood types' throughout this manuscript, but the authors do not state explicitly which flood types are included in the analysis. The authors need to make clear whether they account for coastal flooding, river flooding, or both. The authors should also clarify how the flood maps applied in this study were constructed, both for current and future climate conditions.

3) The authors mention "future population projections", but only provide results for SSP2. The authors need to clarify why they only use one SSP, and why specifically SSP2 was chosen for further analysis. Including multiple SSPs accounts for uncertainty in population projections.

4) The authors poorly position their research in literature on future population exposure to flood hazard. Some "definitions of population at risk" are mentioned, but these only pertain to exposure to coastal flood hazard. The results presented in this study contradict findings of for example Jongman et al. (2012), Merkens et al. (2018), and Neumann et al. (2015), who find that populations will continue to grow in areas of high flood risk. The authors could improve their discussion by reflecting on why their population estimates diverge.

5) There is no mentioning of current flood protection standards nor future adaptation to increasing flood hazard.

Specific/detailed comments:

Abstract

Although climate consequences are mentioned in the title of the manuscript, the abstract appears to only focus on historic and current flood risk. Furthermore, no climate consequences other than increased flood risk are addressed in this study. The description of key findings remains vague, for example, which climate change scenario did the authors use for the reported values? What are areas of disproportionate risk?

Introduction

The overall structure of this section at times is unclear. The goal and methods of the study are mentioned twice: once at "To better understand the impacts...", and under the subsection of "Purpose". Furthermore, results are mentioned in the middle of the introduction ("Results of our research..."), while these can also be read in the abstract and conclusion.

- This introduction is quite long, and at times seems unfocused. For example, the mentioning of extreme weather events is unrelated to the research questions posed in this study.
- The term climate migration is applied here to describe migration driven by increases in flood risk; this should be stated explicitly as other forms of climate migration exist.
- When referring to past research ("Past research on climate migration...") it would be appropriate to refer to this past research. Does this research only relate to flood risk related migration?
- From this introduction it appears that nuisance and episodic flooding will be addressed in this research. However, the researchers only include flooding from return periods of 1 in 5 years.
- Adaptation to increasing flood risk by households and governments is not addressed in this introduction, while its effect on flood risk is considerable (e.g. see de Ruig et al., 2022). It could be mentioned as one of the ways people "respond ... to the same experience of impacts"
- Some sentences could be rephrased to be more concise e.g., "the significant portion of migration taking place at smaller scales", "several types of flooding", "people respond in various ways"
- SSP projections are mentioned but not introduced.
- How will climate change affect coastal flood hazard? How is climate change addressed in this study? Does the research focus on coastal, river, or pluvial flooding? The authors could make this clearer in the last paragraph of the introduction.

Data/ Methods

This section could be hard to follow for those unfamiliar with the R packages used in this study. It furthermore lacks an explanation on how climate change is included in the future population projections (if included at all).

Step 1

- The authors write that:

"In addition to the historical flood exposure data, the FSF-FM current (as of 2020) risk levels were also included at the 2-, 5-, 20-, 100-, and 500-year return periods to account for the proportion of properties in each Census Block are risk of flooding in each of those probabilistic scenarios".

It is unclear what the differences are between historic and current risk levels. Are historic risk levels expressed in expected annual damages? Are they different because of sea level rise/ climate change?

- The abbreviation "FSF-FM" is not introduced.

- A theoretical framework of risk is not provided, the terms "risk" and "exposure" appear to be used interchangeably. Risk could be introduced following a framework, such as described by Ward et al. (2013).

- The term "Multi-Deprivation Index" is not explained. How is this index scored, does it correlate to income and unemployment rates?

- The definition of areas of "considerable flooding" should be made more concise. Why did the authors choose for "~3 times the mean level of flooding"? The meaning of "level of flooding" also remains unclear, does it pertain to flood risk or exposure?

- The goal of the propensity score matching could be made clearer. The "treatment" and "control" are defined using an algorithm, but the authors mentioned before that this grouping is based on "flood levels".

Step 2

- The mentioning of "different types of flood events" in Step 2 is confusing. Does this pertain to pluvial/ fluvial/ storm surges, or to different return periods?

- The variable name "Future change" is confusing as it does not capture any future changes, but instead refers to all unobserved variables ("aimed at capturing any other characteristics of an area which facilitate certain growth rates")

- "perceptions on an area's suitability for raising a family" are mentioned, but there is no mention on perceptions of flood risk itself. Literature indicates that perceptions of risk in part determine the behavior of exposed population (e.g. Becker et al., 2014)

- When selecting return periods of 5, 20, 100, and 500 years a flood protection standard of 5 years is assumed. However, this is not mentioned in the text, and arguments for selecting this standard are not provided.

- "Additional Flood variables available in the full model include the square of the proportion of inundated properties for all return periods". Could the authors explain why this is included next to the minimal model requirements of "proportions of inundated properties for 5, 20, 100, and 500 year floods"?

Step 3

- The authors mention that they develop "climate adjusted population projections" while accounting for flood risk. How were the initial populations adjusted to a future climate, and was the impact of flood risk under climate change already accounted for in these projections? This should be elaborated.

- The authors mention "future flood layer projections". The source of these flood maps is not mentioned, and no description is provided on how these maps were constructed.

- Which climate change scenario is used, and are they coupled to the SSPs assessed in this study?

- Why did the authors choose the SSP2 scenario? Could the authors provide a description of this scenario?

Results

This section in general is too long. Only the most important model results should be discussed here, other results could be moved to the supplement. References to figures should be more concise ("in the image", "Panel A in the figure" ...)

- The authors mention "climate consequences from the historic model results", are results of flood risk under climate change not provided here?

- The "three different indicators" in table 4 could be mentioned in the text for readability.

- The authors propose two reasons why "Ohio actually shows a decrease in population in areas which are less vulnerable ... and an increase in areas that are at risk", but do not provide any literature to substantiate this claim. This would perhaps be more appropriate to move to the discussion section of

the paper.

- The authors mention that "riverine and rainfall risk in these particular inland regions tends to be very concentrated and at a less frequent level than the 5 and 20 year return periods". However, this statement appears to be based on the "historic model results". Perhaps a discussion of the results under the "future flood layer projections" would be more appropriate.
- "...states like North Carolina with exceptional risk in areas like the barrier islands are already starting to see retreat bear out significantly in the historic models, which is further exacerbated by sea-level rise and the increasingly latitudinal reach of cyclonic activity". From the title of the manuscript, it appears that future climate conditions are included in this study. However, results under future climate data are not presented.
- The quality of figures 3 and 4 should be improved. There appears to be no scale on the x-axis, and the labelling of "treatment classes" could be improved to benefit interpretation of the figure. Figures all miss descriptive captions.
- Why is the projected change only shown as "Gain" and "Loss" in figure 5? It would be interesting to further quantify these figures, as providing population development under climate change is the main result of this study.

Discussion

- This section lacks a reflection of the research findings in the context of existing research on migration induced by flood risk in the US (e.g., Hauer (2017)) and research on future population development and flood risk (e.g., Jongman et al. (2012), Merkens et al. (2018), and Neumann et al. (2015)).
- There is no mention of in situ adaptation in this study, whereas literature suggest that migration is only one of many adaptation strategies in face of increasing flood risk (Black et al., 2011; de Ruig et al., 2022). Furthermore, the construction of flood protection infrastructure may mitigate future changes in flood risk. It would be interesting to reflect on this in the discussion of the results.
- The authors mention "this research has been limited to single events and investigations of singular types of flooding (such as nuisance coastal flooding's relationship with property valuation within small neighborhoods)". However, coastal nuisance flooding is never mentioned in the methodology or result section.
- The authors mention that "To our knowledge, there has not been previous research which models population responses for several flood types and projects those relationships into the future with climate change at a high resolution." Current development in agent based modelling is moving towards providing household level simulations of migration decisions in face of climate change on national scales (e.g. Bell et al., 2021; Tierolf et al., 2023). The authors could reflect on how their study contributes/ outperforms such approaches.
- Model validation is included in the supplement but is nowhere discussed. It would benefit the quality of the work to provide a discussion on model performance.

References

- Becker, G., Aerts, J. C. J. H., & Huitema, D. (2014). Influence of flood risk perception and other factors on risk-reducing behaviour: A survey of municipalities along the Rhine. *Journal of Flood Risk Management*, 7(1), 16–30. <https://doi.org/10.1111/jfr3.12025>
- Bell, A. R., Wrathall, D. J., Mueller, V., Chen, J., Oppenheimer, M., Hauer, M., Adams, H., Kulp, S., Clark, P. U., Fussell, E., Magliocca, N., Xiao, T., Gilmore, E. A., Abel, K., Call, M., & Slangen, A. B. A. (2021). Migration towards Bangladesh coastlines projected to increase with sea-level rise through 2100. *Environmental Research Letters*, 16(2), 024045. <https://doi.org/10.1088/1748-9326/abdc5b>
- Black, R., Bennett, S. R. G., Thomas, S. M., & Beddington, J. R. (2011). Migration as adaptation. *Nature*. <https://doi.org/10.1038/478477a>
- de Ruig, L. T., Haer, T., de Moel, H., Brody, S. D., Botzen, W. J. W., Czajkowski, J., & Aerts, J. C. J. H. (2022). How the USA can benefit from risk-based premiums combined with flood protection. *Nature Climate Change*, 1–4. <https://doi.org/10.1038/s41558-022-01501-7>
- Hauer, M. E. (2017). Migration induced by sea-level rise could reshape the US population landscape. *Nature Climate Change*. <https://doi.org/10.1038/nclimate3271>
- Jongman, B., Ward, P. J., & Aerts, J. C. J. H. (2012). Global exposure to river and coastal flooding:

Long term trends and changes. *Global Environmental Change*, 22(4), 823–835.

Merkens, J. L., Lincke, D., Hinkel, J., Brown, S., & Vafeidis, A. T. (2018). Regionalisation of population growth projections in coastal exposure analysis. *Climatic Change*, 151(3–4), 413–426. <https://doi.org/10.1007/S10584-018-2334-8/FIGURES/3>

Neumann, B., Vafeidis, A. T., Zimmermann, J., & Nicholls, R. J. (2015). Future coastal population growth and exposure to sea-level rise and coastal flooding—A global assessment. *PLoS ONE*. <https://doi.org/10.1371/journal.pone.0118571>

Tierolf, L., Haer, T., Botzen, W. J. W., de Bruijn, J. A., Ton, M. J., Reimann, L., & Aerts, J. C. J. H. (2023). A coupled agent-based model for France for simulating adaptation and migration decisions under future coastal flood risk. *Scientific Reports*, 13(1), Article 1. <https://doi.org/10.1038/s41598-023-31351-y>

Ward, P. J., Jongman, B., Weiland, F. S., Bouwman, A., Beek, R. van, Bierkens, M. F. P., Ligtvoet, W., & Winsemius, H. C. (2013). Assessing flood risk at the global scale: Model setup, results, and sensitivity. *Environmental Research Letters*, 8(4), 044019. <https://doi.org/10.1088/1748-9326/8/4/044019>

Reviewer #3 (Remarks to the Author):

The authors used a retrospective dataset of flood exposure at census block level and the most current flood inundation maps for different exceedance probabilities developed by First Street Foundation to calibrate a simple statistic model of population change. The model was used to forecast population changes for the next 30 years across CONUS.

The study is a significant contribution to the field. The recently developed FSF-FM dataset allows to account for different flood types, which is a limitation of previous research where focus was given to specific types (i.e., riverine, pluvial, coastal). The use of high-resolution spatial scale at the scale of census blocks also constitutes an advancement to the state of the art compared to past research that was limited to county or state level. I do not find flaws in the methodology proposed. On the contrary, I find it robust and comprehensive, compared to other studies.

I think the content of the manuscript is a good fit to be published in *Nature Communications*, after addressing minor suggestions.

I find parts of the manuscript difficult to read and follow, especially the paragraphs describing the statistical methods. I also find hard to understand figures 3 and 4. The figures should be standalone; the use of 'treatment' term in legend and axis makes it difficult to digest ; the units / ticks of the x-axis are not clear.

The paragraph describing the challenges of the current framework for estimating population migration due to climate affectation explains clearly the weaknesses of the methodologies used in the literature in previous studies, and explains straightforward the novelty of the proposed method. However, I find a bit confusing for readers that the second paragraph of section 'Challenges with the current framework for linking population change to climate exposure' defines multiple 'flooding types' as the flood probability of exceedance, but the third paragraph uses the term 'types of flooding' for referring to coastal flooding.

In the paragraph 'Future National Level Population Projections Adjusted for Historic Climate Impact' is used a common misconception of the concept of flood return periods, which is described in the

manuscript as a flood level occurring once every x years. Please see this for clarification (<https://www.gfdrr.org/en/100-year-flood#:~:text=The most common misconception is,century%2C> or even this year.

Finally, I think the manuscript would benefit from including a very short discussion on the propagation of the uncertainty in data sources (e.g., adjustment of extreme values for different return periods, errors in hydrodynamic models, and generally, errors in data sources listed in table 1) to the population growth estimates.

REVIEWER COMMENTS

Reviewer #1 (Remarks to the Author):

In this paper, the authors start by developing a “hyper local” model of the association between historical and current flood risk and exposure and population change in the United States, followed by using the estimated coefficients to project population change out to 2055. One of the main findings overall is that the association between flooding and population change is slightly negative (i.e., more/higher flooding is associated with population decline); however, as the authors rightly point out, this finding obscures significant local heterogeneity. In what follows, I provide my major comments, questions, and concerns in an effort to help the authors improve this paper and, ultimately, their contribution to the literature.

In general, I thought the framing of this paper was well organized and complete with the exception of at least two trouble spots. One, at the top of p. 4, is that the authors need to more clearly introduce, discuss, and distinguish the concepts of exposure and risk, as well other related concepts such as vulnerability. The second, which occurs at a couple of points, is that the authors sometimes vacillate between saying “migration” and “population change.” While, in an ideal world, they would have the ability to focus in on migration, in the actual world, they can only get at population change. So, more care is needed not only with respect to consistent word choice, but also with bringing in consideration of the other two components of population change—namely, fertility and mortality.

- This is an excellent point, and one that we agree should be presented more carefully in our research, as well as in academic literature in general. The use of both exposure and risk are the result of the collaborative process and us not explicitly stating a common language for this discussion of the research being undertaken. We have remedied that and are focusing the discussion on the term “exposure” for a couple of different reasons. Most importantly, while the definition of risk requires exposure, it also requires the inclusion of vulnerability. One may argue that vulnerability is accounted for by the measurement of relative inundation proportions within the block groups, where high proportions of inundated properties likely reduces a community’s ability to access resources and infrastructure, and is likely to be correlated with higher damage amounts. As the goal of this manuscript is not to argue that these types of metrics are adequate to capture risk, we have substituted the use of “risk” and replaced it with “exposure” throughout the manuscript. However, with that in mind, the research also accounts for economic and demographic factors, which do capture components of “risk” and vulnerability. As such, we retain some use of the term “risk” when the context is appropriate. The concepts of exposure and vulnerability as components of risk have been explicitly clarified within the Introduction section of the manuscript (1). Additionally, discussion on the use of exposure and risk as used throughout the research is also now included at the beginning of the Data and Methods section (2).
 - (1) *“To understand climate risk, it is important to clarify the definitions of risk components: hazard, exposure, and vulnerability. In this context, hazard refers to*

the probability and severity of physical environmental events. Exposure is concerned with the physical proximity to those hazards, capturing the degree to which assets, people, and systems are exposed. Finally, vulnerability refers to the susceptibility of an exposed system to the impacts of a hazard, often encompassing various social, economic, or political factors that influence the ability to cope with a hazard event (Cardona et al., 2012; Ward et al., 2013). Below, the focus of this research highlights that a flood exposure metric is the primary differentiating factor used in the modeling framework.”

- (2) *“As referenced above, risk is made up of hazard, exposure, and vulnerability. As this manuscript uses the proportion of inundated properties, we refer to the flood information as “exposure” for the remainder of the manuscript. More broadly, other variables which are accounted for in this research include demographic, economic, and political factors, which capture additional aspects of vulnerability, thus constituting the necessary components of risk (Cardona et al., 2012). These additional vulnerability-related metrics are not operationalized in such a way to provide a uniform comparison of “risk” across areas (such as by constructing a risk index), so the use of the term “risk” is kept only in broad contexts.”*

- The reviewer is correct in pointing out that while this paper aims to capture the effects of migration, as migration data is not accessible for the country at a high resolution, we proxy this with population change, or population redistribution over the time period included in the analysis. Therefore, with migration there are additional effects on other population change components (fertility and mortality). These components are captured through the use of demographic and spatial controls, and this has been clarified within the manuscript.
 - *“This research is explicitly focused on the migration component of population change and has taken care to include covariates known to be related to differential levels of fertility and mortality in the research design, captured by social, economic, and spatial factors. Additionally, the research design explicitly removes any changes in fertility and mortality by focusing only on 2 decades of data.”*

Another question that arose for me throughout this paper is how to think about the authors’ “hyper local” approach in light of their reliance on estimating average effects that are then organized at the level of states. For instance, on p. 9, the authors state, “there are overall negative impacts on population growth in areas vulnerable to flood risk,” However, upon reading this, I caught myself wondering whether an overall negative effect really means anything, especially in light of some of the authors’ later comments about spatial heterogeneity and maps such as those in Figure 7. To the authors credit, the authors do dig into specific cases such as Miami-Dade County, FL. However, in doing so, I think they reinforce the false choice between taking an average effects one-size-fits-all approach and a case specific approach. As a middle

ground, I suggest that the authors consider splitting the difference and taking a comparative case study approach that balance case-specific heterogeneity and causal attribution.

- We appreciate the guidance here from the reviewer. As a research team, we went back and forth on this for a considerable amount of time. On the one hand, the ability to bring in additional case studies would highlight the ways in which this process operated in different areas impacted by differential exposure and in different contexts. However, on the other hand, that approach would move us away from the primary messaging we are trying to make around the fact that these relationships occur differently at different scales. We also agree that the primary patterns that exist at the nation, state, and county levels act to hide the hyper-local variation that occurs at the block level, but we believe that is the primary story. In fact, the Miami-Dade story is indicative of what we find in other areas where locations with higher levels of exposure see slower rates of growth and/or losses in population relative to areas with less exposure. We have ultimately chosen to focus on that as the primary narrative tying the results together and have moved some of the results to the Appendix in order to provide enough space for that narrative. Additionally, an inland flooding case example has been added in the Appendix, with reference within the main body of the manuscript.

Reviewer #2 (Remarks to the Author):

The manuscript “Integrating Climate Consequences from Flood Risk into Future Population Projections” presents a modelling study estimating population levels in 2055, whilst accounting for exposure to flood hazard. The methodology presented in this study is applied to census blocks within the contiguous United States. The study first assesses the effect of current flood risk on population development from 2000 to 2020. These findings are then used to adjust a population projection under Shared Socioeconomic Pathway 2 (SSP2). The study finds that current flood risk affects population development in areas of high exposure, resulting in uneven growth in for example Texas, Florida, and California. Population growth is expected to decline and reverse in areas at disproportionate risk within the coming 30 years.

This work presents interesting findings and could be a relevant contribution to the literature on this topic. However, the several issues need to be addressed for the research to qualify for publication.

General comments:

1) Although the researchers mention “climate consequences” the manuscript title, the research only uses historic flood hazard and does not account for changes in flood risk under climate change. Assuming flood hazard will remain the same in the future is an assumption that should be stated explicitly. If the authors do account for climate change, the authors should specifically state which climate change scenarios (or RCP) are analyzed.

- Thank you for making this point, we have clearly gotten too comfortable with the data and process, and did not make this point explicit. Climate change is accounted for within our changing flood model inputs. Flood data is provided from the First Street Foundation Flood Model (FSF-FM), which models flooding from pluvial, fluvial, and coastal sources at several return periods for the current year and projected 30 years into the future under RCP4.5/SSP2. This has been clarified within the manuscript at the first mention of the FSF-FM and within the abstract/introduction.
 - *“...the First Street Foundation Flood Model (FSF-FM) current exposure levels were also included at the 5-, 20-, 100-, and 500-year return periods, which accounts for fluvial, pluvial, and coastal sources, to account for the proportion of properties in each Census Block that are exposed to flooding in each of those probabilistic scenarios. Future flood information is also provided by the FSF-FM, which is projected 30 years into the future under RCP 4.5 using a combination of 21 global climate models, holding all else constant except for climatic changes (for more information, see First Street Foundation, 2020; Bates et al. 2021).”*

2) The authors mention the use of ‘several flood types’ throughout this manuscript, but the authors do not state explicitly which flood types are included in the analysis. The authors need to make clear whether they account for coastal flooding, river flooding, or both. The authors should also clarify how the flood maps applied in this study were constructed, both for current and future climate conditions.

- The flood model outputs utilized in this study account for flooding from pluvial, fluvial, and coastal sources. The flood outputs from the FSF-FM are provided at a 3 meter resolution across the United States, from which this study constructs a block measure of the proportion of properties inundated within various return periods. This has been clarified within the manuscript. To ensure this information on the flood model construction is accessible, we further refer readers to the documentation concerning the FSM-FM (bibliographic references included here for convenience).
 - First Street Foundation. First Street Foundation Technical Documentation. 2020. Available online: https://assets.firststreet.org/uploads/2020/06/FSF_Flood_Model_Technical_Documentation.pdf.
 - Bates, P. D., Quinn, N., Sampson, C., Smith, A., Wing, O., Sosa, J., ... & Krajewski, W. F. (2021). Combined modeling of US fluvial, pluvial, and coastal flood hazard under current and future climates. *Water Resources Research*, 57(2), e2020WR028673.

3) The authors mention “future population projections”, but only provide results for SSP2. The authors need to clarify why they only use one SSP, and why specifically SSP2 was chosen for further analysis. Including multiple SSPs accounts for uncertainty in population projections.

- The future baseline projections are provided at all SSPs. However, the flood data that we utilize in this study (chosen for the high resolution, integration of multiple flooding types,

climate adjustments for future projections, and output of several return periods) provides future projections only for RCP4.5, which is consistent with the SSP2 pathway. As such, for the purposes of this manuscript, to maintain consistency, we project future populations under SSP2 only. We have added information directly into the manuscript (see passage pasted below) to clarify this point and to highlight the fact that this analysis is reflective of both the climate adjusted flood exposure and population projections along the RCP4.5/SSP2 pathway. Future research should incorporate more adjustments on both the climate hazard (flood) and the population projections to better understand the uncertainty which exists from the RCP and SSP trajectories.

- *“While future baseline population projections are provided along all five SSPs, this study utilizes SSP2 for consistency with the flood data (RCP4.5). SSP2 is defined as the “Middle of the Road” scenario, where future development trends are similar to historical patterns, and there is slow progress in achieving sustainable goals (O’Neill et al., 2017).”*

4) The authors poorly position their research in literature on future population exposure to flood hazard. Some “definitions of population at risk” are mentioned, but these only pertain to exposure to coastal flood hazard. The results presented in this study contradict findings of for example Jongman et al. (2012), Merkens et al. (2018), and Neumann et al. (2015), who find that populations will continue to grow in areas of high flood risk. The authors could improve their discussion by reflecting on why their population estimates diverge.

- The reviewer is correct to point out that the provided “definitions of populations at risk” pertain to exposure to coastal flood hazard. While we would like to discuss this point as related to other sources of flood hazard, there is a lack of literature which uses projections of flooding from pluvial and fluvial sources. This is discussed in the preceding paragraph (previously this was the following paragraph, but has been reordered to better provide context).
- Thank you for providing these citations. We agree that our work could be made more clear by situating it within a larger body of literature. In particular, many researchers continue to show that population growth is likely to continue in places with high levels of risk. This is not something that we disagree with, nor do our findings, but we find that growth happens differentially and in areas that see relatively less risk within the same housing markets. Our case study of Miami-Dade, FL highlights this. While the county itself has continued to see growth, the rates have been differentiated between areas that have lost populations, or grown at a slower rate, due to high levels of exposure versus other areas in the housing market which have grown at a faster rate and have lower rates of exposure. At the same level of analysis, our findings are consistent with this research, but our explicit goal was to drill down to lower levels of population aggregation and with high resolution exposure indicators in order to suss out more nuanced patterns. Additionally we come to slightly different conclusions than some of the past research because, most climate migration research investigating flooding risk primarily looks at coastal and hurricane driven flooding, whereas our research is incorporating a higher resolution flood model which also includes rainfall and river flooding. This allows for a

more in depth understanding of what is happening inland and in areas not directly impacted by Sea Level Rise and coastal hurricane flooding. To highlight this point, the suggested papers (and others) have been added into the discussion with further reflections on why these estimates differ (summarized below for convenience).

- *The Jongman et al. paper looks at exposure based on multiplying existing populations by overall future population growth rates. This paper does not look at climate migration or how there may be localized changes in population growth rates with increased flood risk.*
- *The Merkens et al. paper looks at population exposure in a similar way as the Jongman et al. paper does. It projects future populations and overlays them with future flood hazard information to estimate exposure. The Merkens et al. paper does not use historical modeling to quantify relationships and then apply them to future growth.*
- *The Neumann et al. paper similarly looks at broad future population trends and how exposure will change as a result of those population trends. It does not consider out-migration. Said in another way, it uses independent projections of both sea level rise flooding and population growth, and does not look at the relationship between the two where flooding may influence population growth.*

5) There is no mentioning of current flood protection standards nor future adaptation to increasing flood hazard.

- This is a really good point and one that we expect to have a direct impact on how communities deal with increasing exposure into the future. Unfortunately, we do not have the ability to project where future adaptation will take place, but we do have current levels accounted for in the creation of the FSF-FM (per the cited documentation). While adaptation is included within the flood modeling process for the current FSF-FM risk layers and is implied in the historic observation, we cannot project it into the future and therefore it is held constant. The results thus indicate an expectation of how population redistribution might play out given only the isolated impact of climate with no future adaptation. However, this is a strong assumption and thus serves as a limitation for the accuracy of these projections. Discussion about future adaptation has been integrated into the manuscript in the Discussion section (passage provided here for convenience).
 - *“While current adaptation is accounted for, this research does not forecast future adaptation which may serve to dampen push factors or exacerbate pull factors. This serves as an additional source of uncertainty in future projections, and provides opportunities for future research.”*

Specific/detailed comments:

Abstract

Although climate consequences are mentioned in the title of the manuscript, the abstract appears to only focus on historic and current flood risk. Furthermore, no climate consequences other than increased flood risk are addressed in this study.

- The use of “climate consequences” is used in this manuscript to refer to changes in population change rates due to associated changes in flood risk. This climate consequence may be positive or negative, where relatively high flood risk and increases in flood risk may see decreases in population growth rates (or even negative), while areas with relatively low flood risk and relatively low changes in flood risk may see an increase in population growth. This has been clarified in the manuscript.
- Furthermore, changes in flood risk are projected into the future within the flood data sources that this manuscript uses, so that changes in flood risk are captured not only in historic relationships but also may be projected with future relationships. Thus the climate consequences serve as observed historic relationships that have been projected into the future based on projected exposure levels that also change along with the climate.

The description of key findings remains vague, for example, which climate change scenario did the authors use for the reported values? What are areas of disproportionate risk?

- Currently, the flood data source this manuscript utilizes only projects along SSP2/RCP4.5, but the baseline population projections exist along all SSPs. To maintain consistency with the flood data, only the baseline population projections of SSP2 are utilized for future estimates. This has been clarified in the manuscript.
- When referring specifically to the flood metric, we have updated throughout the manuscript to use “exposure” rather than “risk”, and only use risk when discussing the overall study. When referring to areas of disproportionate exposure, the focus is on areas which have very high (relatively) proportions of properties exposed to the underlying flood hazard layer (FSF-FM). This has been clarified and made more explicit within the manuscript. Furthermore, where “disproportionate risk” was previously used, by instead referring to it as “relatively high exposure”.

Introduction

The overall structure of this section at times is unclear. The goal and methods of the study are mentioned twice: once at “To better understand the impacts...”, and under the subsection of “Purpose”. Furthermore, results are mentioned in the middle of the introduction (“Results of our research...”), while these can also be read in the abstract and conclusion.

- We appreciate this point. Upon further reading we could immediately see the redundancy and this helped with cleaning the manuscript so that we could hit the word count limit provided by the journal. Specially, the redundancy in the goals and methods discussion within the Introduction has been reduced by prioritizing the “Purpose”

subsection and removing the earlier mention.

- Additionally, the introduction has been reorganized to strengthen the structure, by removing the first mention of the research goals and methods, synthesizing the repeated materials, and removing the results from the introduction (the only early reference is now in the abstract).
- This introduction is quite long, and at times seems unfocussed. For example, the mentioning of extreme weather events is unrelated to the research questions posed in this study.
 - Please see response above for general focus of introduction. While the authors agree that extreme weather events are not directly related to the research questions posed here, we believe that it serves to contextualize the climate change risk problem. This weather section has been shortened to provide better focus, but as it is only mentioned briefly we do not believe that it warrants removal. We hope that this allows for us to both introduce the context while also removing some of the superfluous and unfocused character of the introduction in the initial submission.
- The term climate migration is applied here to describe migration driven by increases in flood risk; this should be stated explicitly as other forms of climate migration exist.
 - The use of “climate migration” has been clarified for when it relates only to population redistribution driven by flood exposure. There remain some uses of the term “climate migration” that do not refer specifically to flooding, but instead refer to migration due to a broader set of hazards and used only to contextualize the emerging trend of populations moving away from extreme climate hazards. While this is really an international occurrence, there are some instances of this occurring within the US. That being said, the focus of the research has been clarified to highlight the fact that we are really modeling population redistribution as a response to flood exposure.
- When referring to past research (“Past research on climate migration...”) it would be appropriate to refer to this past research. Does this research only relate to flood risk related migration?
 - Thank you for pointing out that we failed to point to the research we were referencing. We have added examples and clarified some of the language as “relatively long-distance”, as we are using this to compare against across blocks (rather than county for example). Additionally, the reviewer is correct to assume that our reference here within the manuscript refers only to flood risk related migration—this has been clarified and is included here for reference.
 - *“Past research on climate migration due to flood risk often focuses on relatively long-distance relocations, such as across county borders (Curtis et al., 2015; Fussell et al., 2010; Gutmann & Field, 2010; Hauer, 2017, Robinson et al., 2020)”*
- From this introduction it appears that nuisance and episodic flooding will be addressed in this research. However, the researchers only include flooding from return periods of 1 in 5 years.

- While data does exist for nuisance flooding (captured as tidal flooding), we were unable to meaningfully isolate it from the pull factors of coastal amenities without significantly reducing the sample size to only coastal regions. That is, as our sample included only of matched block groups which were similar in amenities except for flood exposure, there were not enough appropriate matches, resulting in a very small sample size. It was deemed that this high-probability exposure level would be captured well in the 5-year flooding return period, which would increase the sample size by a significant amount. Furthermore, the inundation from nuisance flooding exists on the very edge of most communities and is more appropriate in the analysis of property-to-property differences. This has been clarified in the manuscript (see below).

- *“These variables exist for various return periods (5, 20, 100, and 500 year return periods) to capture how populations may react differently for different depths and probabilities of flood events, where the reactions in settlement patterns after low-severity and high-frequency flood events may differ from reactions to high-severity and low-frequency flood events. While data from the FSF-FM does include 2 year return period flooding (due to tidal flooding), it is difficult to meaningfully isolate it from other coastal amenities without significantly reducing the sample size. Furthermore, this flooding exists on the edge of communities and is likely more appropriate for property-to-property models.”*

- Adaptation to increasing flood risk by households and governments is not addressed in this introduction, while its effect on flood risk is considerable (e.g. see de Ruig et al., 2022). It could be mentioned as one of the ways people “respond ... to the same experience of impacts”

- While personal adaptation has become more common in high risk areas and is a potential way in which homeowners can mitigate their own personal flood risk (Ruig et al., 2022), other research has also shown that community flooding can be a bigger driver of relocation decisions than an individual’s property risk (McAlpine & Porter, 2018; Keenan et al., 2018). Existing adaptation is built into the flood model, but is held constant into the future. Short discussion on adaptation has been added to both the introduction and discussion (as a limitation and source of uncertainty) sections of the manuscript (highlighted below).

- Introduction: *“Within the United States, research which integrates national scale flood information is often limited to historical records or flood maps provided by government agencies, while high-resolution flood risk information is often only locally available. Additionally, the effect of adaptation on flood risk is considerable (Ruig et al., 2022), and built adaptation should be accounted for.”*
- Discussion: *“While current adaptation is accounted for, this research does not forecast future adaptation which may serve to dampen push factors or exacerbate pull factors. This serves as an additional source of uncertainty in future projections, and provides opportunities for future research.”*

- Some sentences could be rephrased to be more concise e.g., “the significant portion of migration taking place at smaller scales”, “several types of flooding”, “people respond in various ways”

- Thank you for this comment, we have edited and revised the entire manuscript for clarity and multiple sections of the introduction have been reworded, and the provided examples have specifically been changed to be more concise.
- SSP projections are mentioned but not introduced.
 - The introduction of the baseline SSP projections have been clarified at the beginning of the Data/Methods section, where we have added a more explicit introduction of the main underlying data sources (see below for convenience).
 - *“Future baseline population projections, which are demographically driven and do not explicitly account for climate change impacts at a local level, are provided by Hauer (2019) and are proportionally downscaled using current block population counts. While future baseline population projections are provided along all five SSPs, this study utilizes SSP2 for consistency with the flood data (RCP4.5). SSP2 is defined as the “Middle of the Road” scenario, where future development trends are similar to historical patterns, and there is slow progress in achieving sustainable goals (O’Neill et al., 2017).”*
- How will climate change affect coastal flood hazard? How is climate change addressed in this study? Does the research focus on coastal, river, or pluvial flooding? The authors could make this clearer in the last paragraph of the introduction.
 - The flood data (from the FSF-FM) used in this study are provided for the current environment and 30 years in the future under RCP4.5. The flood data includes pluvial, fluvial, and coastal sources, and characterizations exist for several return periods. This has been clarified and better introduced within the manuscript. The specific relationship between climate change and coastal flood hazard, while relevant to this research, is not the focus of this manuscript. This change however, is captured in the flood layers utilized in this study (the FSF-FM). As such, readers are referred to the original research for further information.
 - The research integrates fluvial, pluvial, and coastal flood sources, for the current environment and 30 years into the future under RCP 4.5. This is clarified in the manuscript within the Introduction and at the beginning of the Data/Methods section, where we have added more explicit description in the text about the main underlying data sources. These clarifications are highlighted here for convenience:
 - *“...the First Street Foundation Flood Model (FSF-FM) current exposure levels were also included at the 5-, 20-, 100-, and 500-year return periods, which accounts for fluvial, pluvial, and coastal sources, to account for the proportion of properties in each Census Block that are exposed to flooding in each of those probabilistic scenarios. Future flood information is also provided by the FSF-FM, which is projected 30 years into the future under RCP 4.5 using a combination of 21 global climate models, holding all else constant except for climatic changes (for more information, see First Street Foundation, 2020; Bates et al. 2021).”*

Data/ Methods

This section could be hard to follow for those unfamiliar with the R packages used in this study. It furthermore lacks an explanation on how climate change is included in the future population projections (if included at all).

- The R packages used in this study have been de-emphasized and their methods have been introduced in a way that explains what they do and why they are used. Excerpts are included here for reference ease:
 - *“To account for potential endogeneity due to factors which may influence “self-selection” into block areas with flood risk, propensity scoring was executed to synthetically create conditions akin to a control-treatment design... A matching algorithm is used (Ho et al., 2011), where matches are selected using nearest neighbor as determined by a generalized linear model.”*
 - *“The model selection process was optimized to select the most efficient model based on AIC. Here, the “minimum” and “full” model specifications were provided, where the model iterates through all possible variations between the two to select the one with the lowest AIC, capturing both goodness-of-fit and model simplicity.”*
- The end of the introduction section previously laid out the three steps of the methodology, which walks through:
 - 1) the propensity score matching
 - 2) machine learning approach to identify the flooding/population relationship
 - 3) the application of these relationships to future baseline population projections using future, climate adjusted flood hazards under RCP 4.5.This has been cleaned up and moved instead to the beginning of the Data/Methods section to provide a better overview of the structure of this section. This allows the introduction section to more directly focus on the scope of the project and not the nitty-gritty details of the analysis.
- Additionally, an introduction of the future baseline population data and future flood data was added to the beginning of the Data/Methods section, as these data serve as the primary data (amongst others, also introduced) driving the analysis in this study.

Step 1:

- The authors write that: “In addition to the historical flood exposure data, the FSF-FM current (as of 2020) risk levels were also included at the 2-, 5-, 20-, 100-, and 500-year return periods to account for the proportion of properties in each Census Block are risk of flooding in each of those probabilistic scenarios”. It is unclear what the differences are between historic and current risk levels. Are historic risk levels expressed in expected annual damages? Are they different because of sea level rise/ climate change?

- Historic flood exposure is expressed through records of past flood events, detailed in Lai et al., (2022) and from the NOAA Storm Events Database. These serve as controls so that changes between historic risk and current exposure may be better isolated. While changes to overall risk may be due to a changing climate, the main concern here is related to the difference between floods which have actually occurred and the probabilities of flooding occurring (that is, an area may have an annual 5% probability of flood exposure, but after a flood event, there is a 100% chance that it did occur). Therefore, the current flood exposure levels are of primary importance for this model, and the historic flood events serve as controls for prior sorting based on differences in experienced risk. This has been elaborated within the manuscript (and is highlighted here).
 - *“Both historic exposure and current estimates of the risk of exposure were included to account for two separate dimensions of risk associated with the experience of flooding and the likelihood that one could flood, thus accounting for prior sorting based on experiences of actual flooding.”*
- The abbreviation “FSF-FM” is not introduced.
 - FSF-FM is an abbreviation for the First Street Foundation Flood Model, which is used for current and future flood exposure data. The first use of the FSF-FM abbreviation is now accompanied by an introduction of the full name, as well as basic background information on the data and model.
- A theoretical framework of risk is not provided, the terms “risk” and “exposure” appear to be used interchangeably. Risk could be introduced following a framework, such as described by Ward et al. (2013).
 - The reviewer is correct to point out that the terms risk and exposure were often used interchangeably throughout the manuscript as a result of not defining these terms as relevant to the work described here. While the definition of risk requires exposure, it also requires the inclusion of vulnerability. One may argue that vulnerability is accounted for by the measurement of relative inundation proportions, where high proportions of inundated properties likely reduces a community’s ability to access resources, infrastructure, and is likely to be correlated with higher damage amounts. However, as the goal of this manuscript is not to argue that these metrics are adequate characterizations of risk, we have substituted the use of “risk” for “exposure” throughout the manuscript when specifically referring to these flood metrics. While these flood metrics may be more appropriately characterized as exposure, the research design also takes into account and controls for economic, social, and demographic factors, which account for the vulnerability component necessary to constitute risk. These additional vulnerability-related metrics are not operationalized in such a way to provide a uniform comparison of “risk” across areas (such as by constructing a risk index), so the use of the term “risk” is kept only in broad contexts. An introduction of these terms, with a reference to Cardona et al., (2012) and Ward et al., (2013) are also provided to clarify how they are used within the context of this paper.

- The term “Multi-Deprivation Index” is not explained. How is this index scored, does it correlate to income and unemployment rates?

- The Multi-Deprivation Index used here comes from the Census Bureau. It consists of six dimensions: standard of living, health, education, economic security, housing quality, and neighborhood quality. A short explanation highlighting this has been added, followed by an inline citation (included below for convenience). For further information on how this index is calculated, we invite readers to reference the methodology cited.
 - *“...the Multi-Deprivation Index (MDI; provided by the Census Bureau, capturing six dimensions of standard of living, health, education, economic security, housing quality, and neighborhood quality; Glassman, 2020),”*

- The definition of areas of “considerable flooding” should be made more concise. Why did the authors choose for “~3 times the mean level of flooding”? The meaning of “level of flooding” also remains unclear, does it pertain to flood risk or exposure?

- ~3 times the mean level of flooding was chosen as it is relatively worse than other areas within the state, and does not overly restrict the sample size. Clarification of this has been added to the manuscript.
 - *“Several options for “treatment” variables were considered, where a binary variable was used to define “considerable flooding” versus “not-considerable flooding” in an area. In this analysis “considerable flooding” is a within-state metric that was derived from sensitivity analyses to identify a census block for the treatment group if that census block has a proportion of inundated properties that is ~3 times the mean proportion of inundated properties within census blocks in the same state... This threshold was selected to retain a large sample size while also prioritizing the inclusion of highly exposed blocks.”*
- The “level of flooding” terminology has been changed for clarity throughout the manuscript to specifically refer to the proportion of inundated properties within a block. Throughout the manuscript we use “exposure” to refer to this specific metric of the proportion inundated.

- The goal of the propensity score matching could be made clearer. The “treatment” and “control” are defined using an algorithm, but the authors mentioned before that this grouping is based on “flood levels”.

- Thank you for pushing us on making this clearer. The goal of propensity score matching has been clarified in the manuscript (to account for endogeneity) within the Introduction and the Data/Methods sections, with the “flood level” grouping explicitly stated. We have provided the clarifying passage here for convenience:
 - In the introduction: *“Given the possible endogeneity of the treatment, we use propensity score matching to balance unobserved covariates that affect both the treatment and the outcome and synthetically create a control-treatment design.”*
 - In the Data/Methods section: *“To account for potential endogeneity due to factors which may influence “self-selection” into block areas with flood risk, propensity scoring was executed to synthetically create conditions akin to a control-treatment design. Several options for “treatment” variables were*

considered, where a binary variable was used to define “considerable flooding” versus “not-considerable flooding” in an area. In this analysis “considerable flooding” is a within-state metric that was derived from sensitivity analyses to identify a census block for the treatment group if that census block has a proportion of inundated properties that is ~3 times the mean proportion of inundated properties within census blocks in the same state. In this way, the metric captures relative flood exposure, and a block in a highly exposed state would need to have significantly higher flood exposure to be considered “at risk”. This threshold was selected to retain a large sample size while also prioritizing the inclusion of highly exposed blocks. Conditional on observable characteristics of the block areas, propensity score matching allows for the synthetic construction of comparable “treatment” and “control” groups within a sample, allowing for a more direct comparison of the impacts from flooding on population trends and reducing selection bias.”

Step 2:

- The mentioning of “different types of flood events” in Step 2 is confusing. Does this pertain to pluvial/ fluvial/ storm surges, or to different return periods?
 - Thank you for pointing this out. An introduction of the flood data used in the manuscript has been added earlier within the manuscript, clarifying that pluvial, fluvial, and coastal flooding are considered. The language in step 2 has been changed from “different types of flood events” to more explicitly “different depths and probabilities of flood events”, and the use of “flood types” has been changed throughout the manuscript in all other previous locations.

- The variable name “Future change” is confusing as it does not capture any future changes, but instead refers to all unobserved variables (“aimed at capturing any other characteristics of an area which facilitate certain growth rates”)
 - The variable “Future change” refers to future population growth rates under the baseline population projections. This is introduced and discussed within the manuscript at all mentions of this variable.

- “perceptions on an area’s suitability for raising a family” are mentioned, but there is no mention on perceptions of flood risk itself. Literature indicates that perceptions of risk in part determine the behavior of exposed population (e.g. Becker et al., 2014)
 - The perceptions of flood risk do play a role, but are inherently difficult to capture as how “risk” is defined varies by place. For example, as discussed in the introduction of the paper, how flood impacts manifest differs by area (where, for example, it may manifest as contaminated drinking water, flooded roads causing poor hospital access, and/or ruined crop harvests). Additionally, perceptions of those same impacts may differ significantly by population, where an area that is relatively young with low health needs is less impacted by poor hospital access than a community with high health needs. Additionally, cultural differences may cause differences in risk adversity. News coverage of flood events may also significantly impact perceptions of flood risk, where the higher

availability of flood event memories result in higher perceptions of flood risk. As these perceptions of flood risk are dynamic and thus very difficult to dynamically capture, this manuscript uses spatial controls, future growth controls, and state-level models to capture these variations in perceptions. This discussion is important and a key part of why we believe that these specifications are necessary to include, and thus has been integrated into the manuscript (highlighted below for convenience).

- *“Both the spatial and future change variables also account for differences in differing valuation views by populations, such as perceptions of flood risk (Becker et al., 2014), suitability of neighborhoods for raising a family, and social connectivity.”*

- When selecting return periods of 5, 20, 100, and 500 years a flood protection standard of 5 years is assumed. However, this is not mentioned in the text, and arguments for selecting this standard are not provided.

- The return periods were chosen based on 1) available data from the FSF-FM and 2) the ability to produce meaningful control-treatment comparison groups. For context, the FSF-FM does include 2 year return period data, but it is only included for a very small proportion of blocks in immediate coastal areas. As such, a decision was made to use the 5 year RP as the highest frequency RP in the analysis. This does indicate that areas that are impacted by only 5 year and not 2 year floods will be lumped together with areas that experience both, but it was a tradeoff that we felt needed to be made to more adequately understand the impact of high frequency flood exposure on population redistribution. To the reviewer’s point, this does mean that the 5-year RP becomes our floor and all results should be understood within that context. To that point, we have added a short limitation statement around our inability to explicitly account for more frequent nuisance tidal flooding events (included below for convenience).

- *“While data from the FSF-FM does include 2-year RP flooding (to measure tidal flooding), it is difficult to meaningfully isolate it from other coastal amenities without significantly reducing the sample size. As such, this modeling approach does not explicitly account for nuisance tidal flooding events, but instead operationalizes frequent events at the 5-year RP.”*

- “Additional Flood variables available in the full model include the square of the proportion of inundated properties for all return periods”. Could the authors explain why this is included next to the minimal model requirements of “proportions of inundated properties for 5, 20, 100, and 500 year floods”?

- At this point in the manuscript, all of the variables for the above equations are introduced and discussed. This has been broken into two separate paragraphs to reduce confusion.
- For many variables, including the flood variables, linear, interaction, and squared terms were included in the full model. This allows for the model to select terms that capture any non-linear relationships. This has been clarified in the manuscript:

- *“For many of these variables, linear, interaction terms, and exponential variations (accounting for any non-linear relationships) were included in the full model selection for the final model to consider. The model selection process was optimized to select the most efficient model based on AIC. Here, the “minimum” and “full” model specifications were provided, where the model iterates through all possible variations between the two to select the one with the lowest AIC, capturing both goodness-of-fit and model simplicity.”*

Step 3:

- The authors mention that they develop “climate adjusted population projections” while accounting for flood risk. How were the initial populations adjusted to a future climate, and was the impact of flood risk under climate change already accounted for in these projections? This should be elaborated.

- The future population projections are demographically driven and do not project and take into account other changing factors in the future (such as environmental, political, economic, etc.). These future total populations are controlled along the various SSPs, which ensure that overall population counts at the national level are consistent with their SSPs, but do not otherwise explicitly take into account future climate change impacts at a localized level. This has been elaborated within the manuscript at the introduction of the baseline population projections (added below for convenience).
 - *“Future baseline population projections, which are demographically driven and do not explicitly account for climate change impacts at a local level, are provided by Hauer (2019) and are proportionally downscaled using current block population counts. This study utilizes SSP2 for consistency with the flood data future projections along RCP4.5 (equivalent to SSP2). SSP2 is defined as the “Middle of the Road” scenario, where future development trends are similar to historical patterns, and there is slow progress in achieving sustainable goals (O’Neill et al., 2017).”*

- The authors mention “future flood layer projections”. The source of these flood maps is not mentioned, and no description is provided on how these maps were constructed.

- The research utilizes flood information from First Street Foundation, which integrates fluvial, pluvial, and coastal flood sources, for the current environment and 30 years into the future under RCP 4.5 using a combination of 21 global climate models. This is clarified in the manuscript at the first mention of the FSF-FM as the flood information for this study, as well as in the last paragraph of the introduction. For further information on the construction of the flood model, readers are referred to the original research for further information. The research is cited in the document and provided here for convenience:
 - First Street Foundation. First Street Foundation Technical Documentation. 2020. Available online:

https://assets.firststreet.org/uploads/2020/06/FSF_Flood_Model_Technical_Documentation.pdf.

- Bates, P. D., Quinn, N., Sampson, C., Smith, A., Wing, O., Sosa, J., ... & Krajewski, W. F. (2021). Combined modeling of US fluvial, pluvial, and coastal flood hazard under current and future climates. *Water Resources Research*, 57(2), e2020WR028673.

- Which climate change scenario is used, and are they coupled to the SSPs assessed in this study?

- The research presented here utilizes flood information from First Street Foundation, which models flooding across the United States within the current environment and 30 years into the future under RCP 4.5. SSP 2 is consistent with RCP 4.5, and thus was chosen for consistency. However, baseline population projections exist along all SSPs and may instead be used. This is clarified within the manuscript.

- Why did the authors choose the SSP2 scenario? Could the authors provide a description of this scenario?

- Baseline population projections exist along all SSPs and may instead be used. However, for this study SSP 2 was chosen for consistency with RCP 4.5 (for which the flood model future projections exist). This is clarified within the manuscript.
- A brief introduction of the SSP2 scenario is provided at the beginning of the Data/Methods section, with the newly added sentence:
 - *“Future baseline population projections, which are demographically driven and do not explicitly account for climate change impacts at a local level, are provided by Hauer (2019) and are proportionally downscaled using current block population counts. This study utilizes SSP2 for consistency with the flood data future projections along RCP4.5 (equivalent to SSP2). SSP2 is defined as the “Middle of the Road” scenario, where future development trends are similar to historical patterns, and there is slow progress in achieving sustainable goals (O’Neill et al., 2017).”*

Results

This section in general is too long. Only the most important model results should be discussed here, other results could be moved to the supplement. References to figures should be more concise (“in the image”, “Panel A in the figure” ...)

- This section has been shortened and secondary results have been moved to the appendix. In particular, the authors deemed that the primary focus of the results should revolve around the fact that trends at the national, state, and county levels all hide underlying hyper-local variations at the block level. As such, general national and county trends highlight the high level patterns and the sub-county case-study of Miami-Dade, FL is used to underscore this issue. All other results have been moved to the appendix..

- The authors mention “climate consequences from the historic model results”, are results of flood risk under climate change not provided here?

- Thank you for pushing us to clarify this. The historic analysis was conducted with the control-treatment groups to understand the impact of exposure (specifically the proportion of properties within a block group that are exposed under different flood scenarios (RPs)) and its statistical relationship to population redistribution, historically. Those relationships (operationalized by the modeled coefficient estimates) were then integrated into the SSP baseline projections to statistically correct the projections based on the future climate exposure. Essentially, we’ve identified historic relationships that have already occurred as our best guess as to how future populations will respond to a similar onset of exposure in the future and corrected the SSP projections based on that. We have worked to make this more clear throughout the manuscript.

- The “three different indicators” in table 4 could be mentioned in the text for readability.

- The authors thank the reviewer for this suggestion. We have ensured that the three different indicators are mentioned within the text, and have been cleaned up for an easier relation to the table (highlighted here for convenience).
 - *“The climate consequence across the country is evaluated in Table 4, where the population projections for blocks which are defined as relatively highly exposed to flooding (in respect to three different indicators, defined in the following sentence) are shown to decrease, or have decreased growth, when compared to their respective baseline projections. Blocks were categorized into dichotomous groups for comparison based on the following definitions: if a block has higher than average proportion of inundated properties in the (1) 5, 20, or 100-year RPs, (2) 20-year RP, and (3) 5-year RP.”*

- The authors propose two reasons why “Ohio actually shows a decrease in population in areas which are less vulnerable ... and an increase in areas that are at risk”, but do not provide any literature to substantiate this claim. This would perhaps be more appropriate to move to the discussion section of the paper.

- This is a secondary and illustrative result of the findings within this paper and does not come from literature. This has been clarified within the manuscript. This discussion of Ohio as an example is to help illustrate how observed reactions differ by state (relating to a previous comment about the importance of flood perceptions), how understanding spatial variability within the state is important, and how there is less exposure to flooding in lower return periods. These points have been emphasized, and a more explicit example of how Ohio differs has been added within the Appendix.

- The authors mention that “riverine and rainfall risk in these particular inland regions tends to be very concentrated and at a less frequent level than the 5 and 20 year return periods”. However, this statement appears to be based on the “historic model results”. Perhaps a discussion of the results under the “future flood layer projections” would be more appropriate.

- We thank the reviewer for pointing out the confusion caused by this discussion, especially in light of the lack of explanation about how future flood projections under

climate change are included in our analysis. We have made an effort to better emphasize throughout the manuscript that the future flood projections are indeed climate adjusted, and that our discussion of the results focus on our application of historically observed relationships to future conditions (using population baseline projections and climate adjusted flood projections). In this specific instance, we have clarified and expanded as such:

- *“Additionally, much of the riverine and rainfall exposure in these particular inland regions tends to be very concentrated and at a less frequent level than the 5 and 20-year RPs, for both the current year and under future climate projections, which drive the largest responses in regards to community level population change. Recent research (Kim et al., 2022) has highlighted that inland flood risk/exposure is increasing. This increase is uneven around the country, but the Midwest, Northeast, and Texas/Louisiana Gulf Coast are likely to be the most impacted by increasing inland flooding driven primarily by the increased likelihood and severity of extreme precipitation events. As such, we expect inland areas to increasingly respond to the higher frequency of flood exposure into the future.”*

- “...states like North Carolina with exceptional risk in areas like the barrier islands are already starting to see retreat bear out significantly in the historic models, which is further exacerbated by sea-level rise and the increasingly latitudinal reach of cyclonic activity”. From the title of the manuscript, it appears that future climate conditions are included in this study. However, results under future climate data are not presented.

- Repasting a response to a similar concern above: The historic analysis was conducted with the control-treatment groups to understand the impact of exposure (specifically the proportion of properties within a block group that are exposed under different flood scenarios (RPs)) and its statistical relationship to population redistribution, historically. Those relationships (operationalized by the modeled coefficient estimates) were then integrated into the SSP baseline projections to statistically correct the projections based on the future climate exposure. Essentially, we’ve identified historic relationships that have already occurred as our best guess as to how future populations will respond to a similar onset of exposure in the future and corrected the SSP projections based on that. We have worked to make this more clear throughout the manuscript. Therefore, all of the final results in the manuscript are future climate adjusted based on historic relationships.

- The quality of figures 3 and 4 should be improved. There appears to be no scale on the x-axis, and the labelling of “treatment classes” could be improved to benefit interpretation of the figure. Figures all miss descriptive captions.

- A more descriptive legend has been added for all figures, which specifies which return period treatment is used as the treatment used for comparison.
- Tick marks on the x axis have been clearly added and represent the absolute difference between future population estimates with the climate consequence and current populations or baseline future population projections.

- Descriptions of figures are mentioned within the body of the text and some additional description has been added for the figure captions to ensure readability.

- Why is the projected change only shown as “Gain” and “Loss” in figure 5? It would be interesting to further quantify these figures, as providing population development under climate change is the main result of this study.

- Figure 5 has been updated using a dichromatic scale with 5 classes on each side to further quantify and illustrate how gains and losses differ between counties.

Discussion

- This section lacks a reflection of the research findings in the context of existing research on migration induced by flood risk in the US (e.g., Hauer (2017)) and research on future population development and flood risk (e.g., Jongman et al. (2012), Merkens et al. (2018), and Neumann et al. (2015)).

- Thank you for noting this issue in the development of this section. The Discussion of results in the context of existing research on future population development and flood risk, referencing the suggested literature, has been updated and we have worked to ensure that a linkage to the previous sections of the manuscript have been included. We believe that the new version of the manuscript flows more naturally and has a cleaner connection between the purpose and how the findings relate to that context. Some excerpts are included here for convenience:
 - *“While there is some understanding that populations respond differently to different severity (depths) and probabilities of flooding, that research has typically been limited to single events and investigations of singular types (sources, severities, or probabilities) of flooding. For example, while Hauer (2017) was one*

of the first contributions towards understanding inland population impacts due to flood-related migration, the source of flood information used was still solely from SLR inundation. Advancements in agent based modeling (ABM) are moving towards modeling household level migration under climate change (e.g. Bell et al., 2021; Tierolf et al., 2023; Entwisle et al., 2016), but it has been limited to flooding increases from SLR and requires strong assumptions and extensive existing research to define inputs (for example, amenity values must be previously defined, and thus in these applications are limited), and are computationally intensive due to the use of individual flood simulations.”

- There is no mention of in situ adaptation in this study, whereas literature suggest that migration is only one of many adaptation strategies in face of increasing flood risk (Black et al., 2011; de Ruig et al., 2022). Furthermore, the construction of flood protection infrastructure may mitigate future changes in flood risk. It would be interesting to reflect on this in the discussion of the results.

- While the flood model that is utilized in this study does take into account currently undertaken adaptation measures, these adaptations are held constant into the future. Of course, it likely will not be the case that no future adaptation measures are implemented, but by holding these measures constant into the future, the results of this manuscript may isolate the impact of climate change driven flood risk. This however is an opportunity for future research, and a discussion of both the future research opportunities and the impact on the results presented here has been integrated into the discussion of the manuscript.

- The authors mention “this research has been limited to single events and investigations of singular types of flooding (such as nuisance coastal flooding’s relationship with property valuation within small neighborhoods)”. However, coastal nuisance flooding is never mentioned in the methodology or result section.

- Coastal nuisance flooding is provided as an example to contrast against high-severity and low-probability flooding. While 2 year return period flooding was considered and tested within the research presented here, there were challenges with creating a large enough sample size that might be better suited to isolate the effect of this high-probability and low-severity flooding. As a result, the 5 year return period flooding is used in this analysis to represent high frequency flood exposure. (Borrowed from one of our responses above:) The return periods were chosen based on 1) available data from the FSF-FM and 2) the ability to produce meaningful control-treatment comparison groups. For context, the FSF-FM does include 2 year return period data, but it is only included for a very small proportion of blocks in immediate coastal areas. As such, a decision was made to use the 5 year RP as the highest frequency RP in the analysis. This does indicate that areas that are impacted by only 5 year and not 2 year floods will be lumped together with areas that experience both, but it was a tradeoff that we felt needed to be made to more adequately understand the impact of high frequency flood exposure on population redistribution. To the reviewer’s point, this does mean that the 5-year RP becomes our floor and all results should be understood within that context. To

that point, we have added a short limitation statement around our inability to explicitly account for more frequent nuisance tidal flooding events (added here for convenience). The reference of concern in this comment has been clarified within the manuscript, and the use of nuisance flooding as an example has been removed to reduce confusion.

- *“While data from the FSF-FM does include 2-year RP flooding (to measure tidal flooding), it is difficult to meaningfully isolate it from other coastal amenities without significantly reducing the sample size. As such, this modeling approach does not explicitly account for nuisance tidal flooding events, but instead operationalizes frequent events at the 5-year RP.”*

- The authors mention that “To our knowledge, there has not been previous research which models population responses for several flood types and projects those relationships into the future with climate change at a high resolution.” Current development in agent based modelling is moving towards providing household level simulations of migration decisions in face of climate change on national scales (e.g. Bell et al., 2021; Tierolf et al., 2023). The authors could reflect on how their study contributes/ outperforms such approaches.

- Thank you for raising these important research contributions. We have integrated both of them into our discussion of ongoing work in this area. In short, our work differs from these in that it is focused on small area population impacts of flood exposure across various RPs using a high resolution, deterministically, modeled flood exposure hazard. In contrast, Bell et al. (2021) focuses on coastal flooding in Bangladesh. While this context is not directly comparable to ours, it does provide very good information on the non-US responses to increasing exposure to high frequency flood events. In the case of Tierolf et al. (2023), this is a really interesting application that provides a comparison point for our analysis in a western context but uses a simulation approach driven by a “gravity” model to redistribute populations. While we think this research can certainly be helpful in understanding our own findings, we have chosen to take a different approach by coupling a physical probabilistic flood hazard model with a cohort-change ratio approach in the population projections. In doing so, we are relying almost explicitly on historic/past observations about what we know about observed exposure and population change, then combining those two elements to statistically understand how those known processes interact with one another to give us an understanding of future population projections. By relying on probabilistic characterizations of flood exposure, we are able to account for pluvial, fluvial, and coastal sources, while sets of individual simulations that capture all sources in the US are not widely available, and do not span across the entire CONUS. Additionally, the modeling setup of amenities in the approach taken in our study allows for researchers to easily add and consider many amenities, and how they may be valued differently by different populations. In contrast, the agent-based modeling described requires extensive existing research on amenity valuation, does not allow for differing valuations by populations, and thus only includes proximity to water resources. We have summarized our understanding of the manuscripts below and integrated both into the discussion as comparison points to better understand what our

model can, and cannot, do.

- Bell et al. (2021):
 - The paper finds that the migration model within Bangladesh does not predict enough coastal flooding to drive populations away. While coastal flooding causes a livelihood transition from agricultural to non-agricultural (relevant for opportunities in this context), there are the highest abundance of livelihood alternatives in coastal cities. This relates to the importance of understanding various push and pull factors, especially when the same source of disamenities (coasts causing flooding) is also a source of amenities (coasts allowing city prosperity). In this context, the push factors discussed are related to loss of livelihood opportunities (which may not be as prevalent as a push factor within the United States) as well as the loss of property/dangers posed by flood exposure. For pull factors, the opportunities for livelihood alternatives from agriculture serve as the largest driver, which is likely not as prevalent as a pull factor in the United States.
- Tierolf et al. (2023):
 - Rather than defining and training model specifications on historical data, this approach defines decisions using utility functions with limited and explicitly specified inputs. Here, the inputs include household wealth, amenity value (seemingly defined with coastal amenities only), household income, expected damage per theoretical event, adaptation costs, prospected income and wealth (in migration destination options), amenity values (in migration destinations), and migration costs. Explicitly specified discount rates, risk aversion values, and risk perceptions are included as well. The specification of these inputs require strong assumptions of valuation and utility that hold constant across agents, assuming that agents are the same in several aspects (for example, all agents have the same functions for cost of migration, value of place attachment, etc.). Additionally, there must be adequate existing research defining each of these.
 - Defines amenity values only as coastal amenities. Additionally and relatedly, the modeling approach requires extensive existing research for defining amenity values. While it seems that other amenity values may be built in, there is significantly less flexibility in accounting for many amenities at once, especially if existing research with relevant external validity specifically defining these values does not already exist.
 - The modeling approach requires many simulations of individual and separate flood events at the property level, which is not widely available at a national level for many other countries (to our knowledge, including the US). In contrast, a probabilistic approach for flood exposure is similar (as it requires simulations to define probabilities of exposure to different depths), but the application of it in this context requires significantly less computational resources and may be more widely available in some countries.

- Model validation is included in the supplement but is nowhere discussed. It would benefit the quality of the work to provide a discussion on model performance.
- Thank you for the suggestion. As the validation is relevant to the methodology described in Step 2 of the Methods, a short discussion has been added within this step section, with reference to the Appendix for further details. It is highlighted here for convenience:
 - *“To evaluate the prediction methodology, the same methodology is used but the data are split into training and testing sets through a stratification process. For each state, 80% of the total sample is used to construct the training sample, and the remaining 20% serves as the testing sample set. To evaluate the accuracy of the trained model in making predictions for the testing dataset, three accuracy metrics are considered, keeping with demography field tradition (Hauer, 2019; Smith et al., 2006; Smith & Tayman, 2003; Booth, 2006). Given the evaluation of these accuracy metrics, as well as the conclusions that they are all absolutely low and don’t seem to have any systematic patterns, the methodology to estimate coefficient estimates was deemed reliably useful in this application. This validation process and the results are discussed in further depth in the Appendix.”*

References

- Becker, G., Aerts, J. C. J. H., & Huitema, D. (2014). Influence of flood risk perception and other factors on risk-reducing behaviour: A survey of municipalities along the Rhine. *Journal of Flood Risk Management*, 7(1), 16–30. <https://doi.org/10.1111/jfr3.12025>
- Bell, A. R., Wrathall, D. J., Mueller, V., Chen, J., Oppenheimer, M., Hauer, M., Adams, H., Kulp, S., Clark, P. U., Fussell, E., Magliocca, N., Xiao, T., Gilmore, E. A., Abel, K., Call, M., & Slangen, A. B. A. (2021). Migration towards Bangladesh coastlines projected to increase with sea-level rise through 2100. *Environmental Research Letters*, 16(2), 024045. <https://doi.org/10.1088/1748-9326/abdc5b>
- Black, R., Bennett, S. R. G., Thomas, S. M., & Beddington, J. R. (2011). Migration as adaptation. *Nature*. <https://doi.org/10.1038/478477a>
- de Ruig, L. T., Haer, T., de Moel, H., Brody, S. D., Botzen, W. J. W., Czajkowski, J., & Aerts, J. C. J. H. (2022). How the USA can benefit from risk-based premiums combined with flood protection. *Nature Climate Change*, 1–4. <https://doi.org/10.1038/s41558-022-01501-7>
- Hauer, M. E. (2017). Migration induced by sea-level rise could reshape the US population landscape. *Nature Climate Change*. <https://doi.org/10.1038/nclimate3271>
- Jongman, B., Ward, P. J., & Aerts, J. C. J. H. (2012). Global exposure to river and coastal flooding: Long term trends and changes. *Global Environmental Change*, 22(4), 823–835.
- Merkens, J. L., Lincke, D., Hinkel, J., Brown, S., & Vafeidis, A. T. (2018). Regionalisation of population growth projections in coastal exposure analysis. *Climatic Change*, 151(3–4), 413–426. <https://doi.org/10.1007/S10584-018-2334-8/FIGURES/3>
- Neumann, B., Vafeidis, A. T., Zimmermann, J., & Nicholls, R. J. (2015). Future coastal population growth and exposure to sea-level rise and coastal flooding—A global assessment. *PLoS ONE*. <https://doi.org/10.1371/journal.pone.0118571>
- Tierolf, L., Haer, T., Botzen, W. J. W., de Bruijn, J. A., Ton, M. J., Reimann, L., & Aerts, J. C. J. H. (2023). A coupled agent-based model for France for simulating adaptation and migration

decisions under future coastal flood risk. *Scientific Reports*, 13(1), Article 1.

<https://doi.org/10.1038/s41598-023-31351-y>

Ward, P. J., Jongman, B., Weiland, F. S., Bouwman, A., Beek, R. van, Bierkens, M. F. P., Ligtoet, W., & Winsemius, H. C. (2013). Assessing flood risk at the global scale: Model setup, results, and sensitivity. *Environmental Research Letters*, 8(4), 044019.

<https://doi.org/10.1088/1748-9326/8/4/044019>

Reviewer #3 (Remarks to the Author):

The authors used a retrospective dataset of flood exposure at census block level and the most current flood inundation maps for different exceedance probabilities developed by First Street Foundation to calibrate a simple statistic model of population change. The model was used to forecast population changes for the next 30 years across CONUS.

The study is a significant contribution to the field. The recently developed FSF-FM dataset allows to account for different flood types, which is a limitation of previous research where focus was given to specific types (i.e., riverine, pluvial, coastal). The use of high-resolution spatial scale at the scale of census blocks also constitutes an advancement to the state of the art compared to past research that was limited to county or state level. I do not find flaws in the methodology proposed. On the contrary, I find it robust and comprehensive, compared to other studies.

I think the content of the manuscript is a good fit to be published in *Nature Communications*, after addressing minor suggestions.

I find parts of the manuscript difficult to read and follow, especially the paragraphs describing the statistical methods.

- The manuscript has had a full revisions pass to improve clarity and remove sections from the main text which may distract the reader. The methods section has been reorganized to provide a better overview of the approach, background on the data and methods, and reduce repetition.

I also find hard to understand figures 3 and 4. The figures should be standalone; the use of 'treatment' term in legend and axis makes it difficult to digest ; the units / ticks of the x-axis are not clear.

- A more descriptive legend has been added for all figures, which specifies which return period treatment is used as the treatment used for comparison.

- Tick marks on the x axis have been clearly added and represent the absolute difference between future population estimates with the climate consequence and current populations or baseline future population projections.

The paragraph describing the challenges of the current framework for estimating population migration due to climate affectation explains clearly the weaknesses of the methodologies used in the literature in previous studies, and explains straightforward the novelty of the proposed method. However, I find a bit confusing for readers that the second paragraph of section 'Challenges with the current framework for linking population change to climate exposure' defines multiple 'flooding types' as the flood probability of exceedance, but the third paragraph uses the term 'types of flooding' for referring to coastal flooding.

- The discussion of flood sources and return periods has been clarified for all instances throughout the manuscript. Rather than using a broad term of "flood types," which may confuse readers, the manuscript now explicitly states flood sources and return periods (probabilities). Example provided below:
 - *"By accounting for population and spatial characteristics, migration likelihoods may be captured in population growth (and decline) rates as associated with various sources, depths, and probabilities of flood exposure."*

In the paragraph 'Future National Level Population Projections Adjusted for Historic Climate Impact' is used a common misconception of the concept of flood return periods, which is described in the manuscript as a flood level occurring once every x years. Please see this for clarification (<https://www.gfdrr.org/en/100-year-flood#:~:text=The most common misconception is,century%2C or even this year.>

- The reviewer is correct to point out that return periods are commonly misunderstood, for example, where many assume that a 20 year return period means that the event will happen only every 20 years. In reality, this refers instead to a 5% annual probability—such an event may occur several times within a 20 year period, or not at all. The language in the manuscript has been updated to not contribute to this confusion.

Finally, I think the manuscript would benefit from including a very short discussion on the propagation of the uncertainty in data sources (e.g., adjustment of extreme values for different return periods, errors in hydrodynamic models, and generally, errors in data sources listed in table 1) to the population growth estimates.

- There may be uncertainty at multiple steps throughout this process. For example, historical data may have uncertainties due to accuracy and measurement issues. In addition, flood hazard data is based on a probabilistic modeling approach. Future data (flood and population) may have additional uncertainties as it builds off of historical data and uses various modeling approaches, capturing climate uncertainties, modeling

uncertainties, and uncertainties due to underlying assumptions. This has been integrated into the manuscript, and is referenced here for convenience.

- *“While current adaptation is accounted for in the FSF-FM used in this analysis, one limitation of the analysis is that it does not forecast future adaptation which may serve to dampen push factors or exacerbate pull factors. This serves as a potential source of uncertainty in future projections, and provides opportunities for future research. Additionally, there may be uncertainty at multiple other steps throughout this process. The first forms of uncertainty come from the underlying flood, population, and other input data. Historical data may have uncertainties due to accuracy and measurement issues, while future data (flood and population) may have additional uncertainties as it builds off of historical data and uses various modeling approaches, capturing climate uncertainties, modeling uncertainties, and uncertainties due to underlying assumptions.”*

REVIEWERS' COMMENTS

Reviewer #1 (Remarks to the Author):

Having twice reviewed this paper, I am satisfied with the authors' efforts to address my earlier concerns.

Reviewer #2 (Remarks to the Author):

The revised version of "Integrating Climate Consequences from Flood Risk into Future Population Projections" has improved considerably compared to the initial version of the manuscript. The authors have clarified their definition of risk throughout the manuscript, elaborated on the climate and socioeconomical scenarios used to project development into the future, and have positioned their research better in current literature. The authors furthermore condensed sections to ease comprehension and remade figures to include more descriptive legends and color scales. I would suggest accepting the manuscript upon revision of one remaining minor comment:

1. In Response 2, the authors state: "The flood outputs from the FSF-FM are provided at a 3 meter resolution across the United States, from which this study constructs a block measure of the proportion of properties inundated within various return periods. This has been clarified within the manuscript." However, it remains unclear where this has been clarified in the manuscript.